# Glass plate sampling efficiency for trace gases in the sea surface microlayer

Lea Lange<sup>1</sup>, Dennis Booge<sup>1</sup>, Hendrik Feil<sup>1,2</sup>, Josefine Karnatz<sup>1</sup>, Ina Stoltenberg<sup>1</sup>, Hermann W. Bange<sup>1</sup>, Christa A. Marandino<sup>1</sup>

<sup>1</sup>Marine Biogeochemistry, GEOMAR Helmholtz Centre for Ocean Research Kiel, 24148, Germany <sup>2</sup>Institute for Chemistry and Biology of the Marine Environment (ICBM), Carl von Ossietzky Universität Oldenburg, Germany *Correspondence to*: Lea Lange (llange@geomar.de)

**Abstract.** Many climate-active trace gases in the atmosphere are closely linked to production and consumption in the ocean, which are, in turn, influenced by the sea surface microlayer (SML). The SML is the upper most layer of the ocean with up to 1 mm thickness, often enriched in organics. Studies of trace gases in the SML aim to identify and quantify potential processes unique to the SML and to understand the SML's influence on the transfer between air and sea. Established sampling techniques of the SML (e.g., glass plate, mesh screen) are associated with high losses for the volatile trace gases. Despite the high losses, in this study we find that meaningful analysis of glass plate samples for trace gases is possible. We experimentally determined the sampling efficiency for the short-lived trace gases dimethyl sulphide (DMS), isoprene, and carbon disulphide (CS<sub>2</sub>). Water temperature and trace gas concentration were the main drivers for sampling efficiency variability, while salinity and the number of dips of the glass plate were not significant. The effect of surfactants could not finally be untangled. Although our results are consistent, we do not quantify a sampling efficiency to correct individual measurements, as our experiments did not encompass the full suite of environmental parameters normally encountered in the field. Instead, we suggest to use  $0.13 \pm 0.01$  ( $\pm$  standard error) for DMS and isoprene, and  $0.12 \pm 0.01$  for CS<sub>2</sub> as thresholds to identify cases of net production in the SML. Future studies should extend to long-lived species (e.g., nitrous oxide, methane), include the effect of wind, and be repeated for the mesh screen. We hypothesize that a correction of individual measurements requires to determine sampling efficiency as a function of environmental parameters, for which the underlying physicochemical relationships need to be unraveled by increasing the parameter space studied here.

# 1 Introduction

Short- and long-lived trace gases impact Earth's climate via processes like hydroxyl radical chemistry, aerosol formation, cloud condensation nuclei formation, or the greenhouse effect. The oceans serve as important sinks and sources for many of these gases. Understanding trace gas cycling in the ocean mixed layer and exchange with the atmosphere across the air-sea interface is, therefore, crucial for climate and air quality predictions. (Liss et al., 2014) Trace gas data from the upper meter is scarce and the lack of information is even worse for the surface microlayer (SML). The SML is the uppermost layer of the water column with a thickness of less than 1 mm. It often shows organic enrichment, for instance with surface-active substances

35

40

(surfactants), especially during calm conditions (Wurl et al., 2011). Enrichment is often indicated as enrichment factor (*EF*), defined as the ratio between the quantity's concentration in the SML and in the ULW. Surfactants alter the physicochemical properties of the SML, like reducing the surface tension, and are addressed in most SML studies. For high concentration of surfactants so-called slicks form that are often visible as flat surfaces, as they dampen the waves. The SML has been studied for a wide range of physical, chemical and biological parameters, which resulted in the hypothesis of the SML being a biogeochemical reactor (Bibi et al., 2025). However, established sampling techniques are not well suited for volatile gases, causing a severe lack of information on trace gas concentrations directly at the air-sea interface. For ease and convenience, many air-sea gas exchange estimates are thus based on underlying water (ULW) concentrations sampled from the mixed layer (usually around 5 m depth). Studies have shown evidence that this neglect of processes occurring shallower than 5 m and in the SML may be the reason for discrepancies between computed fluxes and predicted or measured fluxes (Booge et al., 2016; Kock et al., 2012; Lennartz et al., 2017; Marandino et al., 2005, 2008). The SML is ubiquitous in the ocean, therefore inaccuracies in air-sea gas exchange estimates may accumulate and may show a significant effect on atmospheric chemistry and climate computations even on a global scale (Sabbaghzadeh et al., 2017; Wurl et al., 2011). To unravel the relevance of the SML on air-sea gas exchange and its potential as a source or sink for trace gases, it is necessary to sample the SML efficiently and accurately.

In order to measure SML concentrations and gradients at sea, samples must be taken by hand with specialized equipment from a small rubber boat, which is less convenient and more difficult than the common underway pumping system or discrete water samples from a CTD rosette used on board during many oceanographic campaigns. A widely used sampling device is the glass plate, which was first described by Harvey and Burzell (1972) for collection of microorganisms and organic content, but other techniques exist as well, such as the mesh screen (Garrett, 1965), rotating drum (Harvey, 1966), membrane sampler (Crow et al., 1975), cryogenic sampler (Turner and Liss, 1985) and more recently a gas-permeable floating tube specially designed for trace gases (Saint-Macary et al., 2023). Several studies discuss the limitations and difficulties to compare results from the different methods for non-volatile compounds (e.g., Cunliffe and Wurl, 2014) and additionally the need for a new technique for volatile compounds like trace gases is highlighted (e.g., Engel et al., 2017). There are only a few studies that sampled the SML for trace gases and even fewer that used the glass plate, although it is the most common device for other parameters measured in the SML. To the best of our knowledge, there are no studies with SML samples for the short-lived trace gas isoprene, and only one on carbon disulphide (CS<sub>2</sub>; Turner and Liss, 1985), but dimethyl sulphide (DMS) measured from SML samples has been used more widely to determine EF (see Saint-Macary et al., 2023; Turner and Liss, 1985; Walker et al., 2016; Yang, 1999; Yang et al., 2001). In those and similar studies, glass plate and mesh screen sampling are usually described as being prone to sampling losses due to volatilization, but to the best of our knowledge the magnitude of the losses has not been tested and no corrections attempted. One study accompanied their field measurements of DMS EF with laboratory work to investigate the dependence of the EF on factors like temperature, salinity, and DMS concentration (Yang et al., 2001). They address potential losses, but their experimental design does not allow to infer reliable estimates of sampling efficiency. Neglecting losses might lead to wrong interpretation of calculated enrichment factors, especially when losses are high.

Trace gas losses in SML sampling are consistently addressed in relevant publications using the glass plate or mesh screen, because the sampled water is spread in a very thin film over a large surface exposed to air before it is transferred to a sample bottle. Diffusive losses driven by the concentration gradient between air and sample water are larger with stronger gradients. Additionally, the exposure to light might drive photochemical processes on the glass plate or mesh screen. Furthermore, the sample bottle is open for the duration of sampling, as usually one dip of the glass plate or mesh screen does not provide enough sample volume for analysis. Finally, the sample is wiped from the glass plate and drips into the sample bottle for both the glass plate and mesh screen, introducing additional turbulence that increases the losses further. Apart from losing gas to the surrounding air, there is also a potential loss source from adsorption to the glass plate and mesh screen surface, the wiper, and the funnel.

Studies using a gas-permeable floating tube (e.g., Saint-Macary et al., 2023) attempt to overcome those issues, but their results are difficult to integrate with quantities obtained from the glass plate or the mesh screen. Results from different SML sampling techniques are in general complex to compare for three reasons: (1) sampling thickness differs significantly between techniques (2) sampling losses vary (volatilization, adsorption, undersize, ...) (3) introduction of error (e.g., stressing phytoplankton by sample handling). Even for non-volatile compounds, like surfactants, sampling losses might not necessarily be negligible (see controversy on mesh screen sampling efficiency in Garrett (1974)), though in general they are considered to be. For these reasons, trace gas concentrations and resulting *EF* vary significantly between SML techniques (see Turner and Liss, 1985; Walker et al., 2016; Yang et al., 2001), which commonly is attributed to differences in sampling thickness.

This study has been conducted in the context of the joint, interdisciplinary campaigns within the project Biogeochemical processes and Air-sea exchange in the Sea-Surface microlayer (BASS), where the glass plate technique was used. To enable the best possible comparability, we aim, in this study, to identify sampling efficiency of the glass plate technique for volatile trace gases like DMS, isoprene and CS<sub>2</sub>. Due to the volatile nature of these gases, sampling losses are expected to be high for glass plate sampling, yet, existing studies have shown that glass plate samples are still above the limit of detection. We therefore hypothesize that meaningful analysis of glass plate samples is possible. Physicochemical properties of the sample, like temperature, salinity, and trace gas concentration in ULW, as well as the individual steps performed in glass plate sampling are expected to affect the sampling loss. As those differ between samples, we expect to see variations of the sampling losses correlated with those parameters. Therefore, in this study we make the first attempt to derive a correction method for the classic glass plate technique. To do so we investigated a range of isotopically labelled atmospherically short-lived trace gases (DMS-d3, isoprene-d5, <sup>13</sup>CS<sub>2</sub>), addressing the following questions: (1) Are SML samples from the glass plate meaningful for trace gases? (2) If so, what drives the associated losses of those volatile compounds? (3) Can we derive a method to assess the SML enrichment accurately in experiments and in the field?

#### 95 2 Methods

100

To quantify the sampling efficiency of the glass plate, one would have to compare in situ concentrations in the SML to the measured concentration from the glass plate sample. However, there is no established sampling technique or in situ sensor for trace gases in the SML. We, therefore, performed experiments in a tank where the water was mixed immediately before sampling. This ensured that the SML had the same concentration of trace gases as the ULW, which can be sampled without losses. The deviation of the concentration measured in the SML sample from the ULW samples reflects the sampling efficiency of the glass plate method. Additionally, the mixing ensures that the SML sample is not diluted by ULW (as  $C_{SML} = C_{ULW}$ ) and we, therefore, capture the pure sampling error. Note, that when we use SML here we refer to the upper 1 mm of the water body, although there is no enrichment of organics present.

## 2.1 Experimental setup

- In 2023, we conducted a series of lab experiments to assess the trace gas loss during glass plate sampling: (A) preliminary test (B) glass plate sampling without surfactants (C) glass plate sampling with surfactants. The preliminary test from experiment A are described in Appendix A Preliminary test, while we exclusively address experiment B and C further on. To exclude biological and chemical consumption or production of gases as potential sources or sinks affecting the measured concentrations, we used isotopically labelled gases in fresh water (FW) and artificial seawater (AS): deuterated DMS (DMS-d3), deuterated isoprene (isoprene-d5) and <sup>13</sup>CS<sub>2</sub> as representatives for DMS, isoprene and CS<sub>2</sub>, respectively. Trace gas concentration, water temperature, salinity, and surfactants were varied in the ULW to assess their effect on sampling efficiency, whereas surfactants in the SML, sampling duration for SML, sample volume of SML samples, number of glass plate dips, pH in ULW, air temperature and person who sampled SML were recorded to accompany our measurements. To exclude sampling bias with glass plate sampling, only one person was sampling SML samples (with exceptions in experiment B). The experiments were performed with different media to address two research objectives:
  - (1) Pure sampling efficiency was determined using ultrapure water (as fresh water) and artificial seawater with varying temperatures, salinities and trace gas concentration, as those parameters affect the solubility and diffusivity of gases.
  - (2) Artificial surfactants were mixed into the artificial seawater in three different amounts to understand if their presence affects the losses.
- Ultrapure water (18.2 M $\Omega$ cm) was provided from lab water purification systems (Arium® Pro, Sartorius, Göttingen, Germany). Artificial seawater was either prepared with Tropic Marin® Pro-Reef Sea Salt (Tropic Marin AG, Hünenberg, Switzerland) or with ordinary aquarium salt in ultrapure water aiming for a practical salinity ( $S_P$ ) above 30.0. Every day we spiked the water with isotopically labelled trace gases. The concentrations of isotopes varied between the experiments, as two different mixtures of standards were used.

Table 1 summarizes the experiments, which took place in two different locations in Germany: (1) under the roof of the SURF facility at ICBM, Wilhelmshaven and (2) in the laboratories at GEOMAR Helmholtz Centre for Ocean Research Kiel. Two different tanks were used: (1) aquarium made of glass (2) "AZ crate" (Polypropylene, Schoeller Allibert, Schwerin, Germany).

Table 1 Overview of experiments performed in this study. Sample volumes are the targeted volumes. Analysis gas chromatography—
mass spectrometry is abbreviated as GC-MS. Research objectives: (1) pure sampling efficiency (2) effect of surfactants on sampling efficiency.

| Experiment                   | Tank                                         | Treatments                                            | Medium                                                                               | Analysis<br>Sample volume      | Additional measurements                                           | Research objectives |  |
|------------------------------|----------------------------------------------|-------------------------------------------------------|--------------------------------------------------------------------------------------|--------------------------------|-------------------------------------------------------------------|---------------------|--|
| A<br>(GEOMAR)<br>13–15 March | incubation<br>bath                           | fresh water                                           | ultrapure water                                                                      | GC–MS<br>10 mL                 |                                                                   | (1)                 |  |
| В                            |                                              | fresh water                                           | ultrapure water<br>63–68 L                                                           |                                |                                                                   |                     |  |
| (ICBM)<br>18–20 April        | aquarium                                     | artificial seawater                                   | Tropic Marin®  Pro-Reef Sea Salt  74 L, $S_P = 34.5$ 7.5 °dKH                        | GC–MS<br>10 mL                 | occasional $T_{water}$ $T_{air} \approx 12.7  ^{\circ}\mathrm{C}$ | (1)                 |  |
| C<br>(GEOMAR)                | fresh water  "AZ crate"  artificial seawater |                                                       | ultrapure water $78 L$ $pH = 6.3$ ultrapure water with                               | GC–MS<br>10 mL                 | $T_{water}$ . $T_{air} pprox 23.1~^{ m o}{ m C}$ .                | (1), (2)            |  |
| 28 July –<br>31 August       | AZ CIAIC                                     | artificial seawater<br>with surfactants<br>(3 levels) | ordinary aquarium salt<br>78–81 L<br>S <sub>P</sub> : 29.1 and 33.9<br>pH: 8.06–7.84 | GC–MS,<br>voltammetry<br>15 mL | $(\Delta T_{air} = 4 \text{ °C})$                                 | (2)                 |  |

# 2.2 Sampling

Before each sampling, we mixed the water body carefully for one min and waited for two min for the turbulent motion to subside. We sampled the SML, ULW (10–13 cm) and (1–6.5 cm). All samples were collected in amber borosilicate glass bottles (20 mL). SML samples in experiment A (preliminary test) were taken with the mesh screen (Garrett, 1965). All other SML samples were taken with the glass plate technique (Harvey and Burzell, 1972). Following the recommendation by

Cunliffe and Wurl (2014) and examples from recent studies (e.g., Adenaya et al., 2021), we reduced the speed of extracting the glass plate to 5 cm/s. The glass plate (borosilicate glass,  $30 \text{ cm} \times 25 \text{ cm} \times 0.5 \text{ cm}$  in experiment B,  $40 \text{ cm} \times 30 \text{ cm} \times 0.5 \text{ cm}$  in experiment C) was immersed to a depth of about 20 or 25 cm. Both sides of the glass plate were wiped with a regular silicone or rubber wiper and the run-off was collected into a sample bottle with a plastic funnel. A single dip provided between 1.8-6.3 mL of medium. To collect sufficient sample volume for analysis two to five dips were required for analysis with gas chromatography—mass spectrometer (GC–MS) and voltammetry samples. The thickness of the sampled layer is determined by Eq. (1) (Cunliffe and Wurl, 2014).

$$h_{SML} \approx h_{sampled} = \frac{V_{sample}}{n_{divs}A}$$
 (1)

where  $h_{SML}$  is the true thickness of the SML,  $h_{sampled}$  is the thickness that was actually sampled in cm,  $V_{sample}$  is the total volume of the glass plate sample in mL, A is the wetted surface area of the glass plate in cm<sup>2</sup> and  $n_{dips}$  is the number of dips that were needed to collect  $V_{sample}$ .

Each SML sample is always associated with a reference ULW sample and sometimes with additional ULW or surface samples. All samples taken together after the same mixing event are referred to as a sample set. The reference ULW sample was taken immediately after the first dip of the SML sample was collected, interrupting the SML sampling for less than a minute (overall SML sampling took 90–235 s). All additional ULW or surface samples were taken after the SML sample had been finished (exception from this order is 19 April in experiment B). ULW samples were taken with a pipette (5 and 10 mL, Eppendorf SE, Hamburg, Germany) set to the required sample volume by inserting the tip of the pipette vertically and carefully into the water. For sample volumes of 15 mL, a pipette was immersed twice in the same location. ULW samples were taken from a water depth below surface of ~10 cm in experiment B and ~13 cm in experiment C. Surface samples were sampled the same way as ULW samples, with the only difference that we were aiming for a depth below surface of less than 1 cm in experiment B and ~4.5 cm in experiment C (with exception of 31 August: ~6.5 cm). Additional ULW and surface samples were collected once or twice daily to confirm that the tank was well mixed. Before each sample set was obtained, all of the equipment was rinsed with ultrapure water.

Samples were taken for trace gas analysis and voltammetry (surfactants). Trace gases were analysed during all of the experiments, whereas voltammetry was only performed in selected cases (Table 1). The targeted sample volume for trace gas analysis was 10 mL. SML samples were filled to at least the targeted sample volume and later on weighed to calculate the sample volume based on the density of the medium. In the treatment with surfactants, a larger sample volume was required for voltammetry measurements to permit a subsequent 10 mL subsample. Given the limited SML volume in the tank and the presence of surfactants, collecting two true replicate SML samples was not feasible. Consequently, the target sample volume was increased to 15 mL for GC–MS analysis, and the purged sample for GC–MS analysis was subsequently reused for surfactant analysis. No effect on final concentrations from the GC–MS measurements was found between 10 mL and 15 mL samples, as gases are stripped completely out of the medium within the purge time. Sample bottles used for the voltammetry measurements were additionally acid-rinsed (10 % HCl) and pre-combusted at 500 °C before sampling.

# 2.3 Sample analysis trace gases

The system used to measure the trace gases DMS-d3, isoprene-d5, and <sup>13</sup>CS<sub>2</sub> is a purge and trap GC-MS (Zavarsky et al., 170 2018). It consists of a self-constructed purge and trap system, a type 7890A GC, and a 5975C MS (Agilent, Waldbronn, Germany). The GC uses a capillary column (Supel-Q-PLOT, 30 m × 0.32 mm, Merck KGaA, Darmstadt, Germany). The MS is equipped with electrical ionisation, a quadrupole mass filter and a triple-axis electron multiplier detector. All three compounds are measured from the same sample. As samples cannot be preserved, no replicates could be taken for GC-MS analysis. Samples were stored at room temperature in the dark and measured within less than one hour after sampling. Samples 175 (both 10 and 15 mL) are purged with helium (99.999 %, Air Liquide, Düsseldorf, Germany) at a flow rate of 80 mL min<sup>-1</sup> for 10 min. The sample gas stream is dried with a Nafion<sup>®</sup> membrane dryer (Perma Pure, Lakewood, United States) and trapped with liquid nitrogen in a Sulfinert® stainless steel tube (Restek GmbH, Bad Homburg, Germany). Injection into the GC is semi-automated with a 6-port valve that channels the carrier gas flow through the trap while it is heated with hot water (70-180 100 °C) to release the sample. The MS is operated in single-ion mode. The peak areas (PA) in the spectra were integrated manually with MSD Productivity ChemStation (version E.02.02.1431, Agilent, Waldbronn, Germany). The unit of PA is arbitrary and, therefore, is not used throughout the manuscript. Qualification and quantification of DMS-d3, isoprene-d5, and <sup>13</sup>CS<sub>2</sub> was done using m/z ratios of 64 and 65, 72 and 73, and 77 and 79, respectively. The limit of detection (LOD, i.e., minimum PA detectable) was defined as 7 times the root mean square error of the baseline noise and was determined for each 185 compound individually from one representative chromatogram for each experiment. This yielded LODs for DMS-d3 of 209 and 90, for isoprene-d5 of 300 and 309, and for <sup>13</sup>CS<sub>2</sub> of 258 and 181 respectively for experiment B and C. To extract only isotope signals, all PA were corrected for spectral fragmentation, which causes overlap in the mass spectra of natural and isotopically labelled compounds. A calibration was deemed not necessary, as we are only interested in ratios of SML trace gas content over ULW trace gas content, each sampled directly one after another. The calibration usually used with this setup 190 translates PA linearly into concentrations, which means that the ratio of PAs equals the ratio of concentrations, thus the ratio of PAs is sufficient. It takes less than 1 h to measure all samples from one set, therefore also the drift of the GC–MS is negligible for one sample set. By skipping the calibration, the number of measured SML-ULW pairs for each day was increased. Each PA was normalized by the sample volume to allow for comparability. Associated uncertainty of GC-MS measurements is 10 %. An additional step in quality control (QC) was added to identify measurements with poor quality. Peaks of mass 91 at 195 retention time  $\sim 8.10$  min were integrated for all SML, ULW and surface samples and normalized by sample volume  $(PA_{91})$ . Mass 91 is associated with toluene, as there is a stable toluene contamination in the purge system. Therefore, the variability in  $PA_{91}$  is indicative of poor-quality measurements (Appendix B Additional quality control using mass 91).

# 2.4 Sample analysis surface activity

200

Surface activity (SA) of surfactants was measured only for samples when Triton X-100 (TX-100, with molecular weight 625 g mol<sup>-1</sup>, Merck, Darmstadt, Germany) was added to the tank (experiment C, see Table 1). Of those samples, only a subset

210

was measured, which was a representative ULW triplicate at the beginning and end of the day, as well as each SML (singlets) sample. SA was quantified using phase-sensitive alternating current voltammetry with a hanging mercury drop electrode 797 VA (Computrace, Metrohm, Switzerland), following the method established by Ćosović and Vojvodić (1982, 1998). This electrochemical technique exploits the adsorption behaviour of surfactants at the interface between the mercury electrode and electrolyte, altering the capacitive current (Scholz, 2015). Samples were stored at -20 °C until analysis. Prior to measurement, all samples were brought to room temperature and adjusted to a uniform ionic strength corresponding to a  $S_P$  of 35.0 (0.55 mol  $L^{-1}$  NaCl) to ensure comparability across measurements. Depending on the TX-100 concentration added to the tank, measurements were performed with a deposition time ranging between 10–60 s and a voltage sweep between -0.6 and -1.0 V. Additionally, high TX-100 concentration samples required dilution before measuring (dilutions from 0.7–0.97). For evaluation, the initial current response at -0.6 V was used, and the difference in capacity current between sample and blank was calculated with  $\Delta I = I_{blank} - I_{sample}$ . Three scans were recorded per replicate and the mean of the three scans was used for final analysis. Calibration was performed using TX-100 across a concentration range of 0.01–1.3 mg  $L^{-1}$ . Only the linear portion of the response curve was used for quantification. Analytical precision was assessed using daily standards and blanks, yielding an average precision of 6 %, with all values remaining below 10 %.

For each triplicate of ULW surfactants samples the standard error (*SE*) of mean was calculated with Eq. (2). Only a few representative ULW samples were taken per day and then were averaged over the day. Errors were propagated with Eq. (3) to show the precision of the daily mean accounting for replicate uncertainty.

$$SE = \frac{SD}{\sqrt{n}} \tag{2}$$

$$SE_{day} = \frac{1}{m} \sqrt{\sum_{i=1}^{m} SE_i^2} \tag{3}$$

where SE and  $SE_i$  are the standard error of the triplicate, SD is standard deviation of the triplicate, n the number of samples per triplicate,  $SE_{day}$  is standard error of the mean of triplicates per day and m is number of triplicates per day.

Spread of triplicate means is calculated as SD of the mean values with Eq. (4).

$$SD_{day} = \sqrt{\frac{1}{m-1} \sum_{i=1}^{m} (\bar{x}_i - \bar{x}_{day})^2}$$
 (4)

where  $SD_{day}$  is standard deviation for the daily mean, m the number of triplicates,  $\bar{x_i}$  the mean of the triplicate and  $\bar{x}_{day}$  the daily mean.

#### 2.5 Sampling efficiency

Sampling efficiency is calculated as ratio of trace gas content per volume in the SML over content per volume in ULW (Eq. (5)). Units should be a concentration (e.g., mol L<sup>-1</sup>) or linearly proportional to the resulting concentration (e.g., PA as per millilitre of sample). Sampling efficiency is given as a dimensionless fraction. The complement of this fraction represents the

sampling losses (Eq. (6)). Note, that this resembles the *EF* commonly computed to determine enrichment in the SML in field samples, e.g.,  $EF = x_{SML}/x_{ULW}$ , where x is a measured property.

$$E = \frac{PA_{SML}}{PA_{ULW}} \propto \frac{C_{SML}}{C_{ULW}} \tag{5}$$

$$L = 1 - E \tag{6}$$

$$C_{SML,true} = \frac{C_{SML}}{E} \tag{7}$$

where E is sampling efficiency (unitless), PA is the peak area per millilitre of sample, C is a concentration in, e.g., mol L<sup>-1</sup>, taken from the SML or ULW, as indicated by the subscripts and L are the sampling losses.

Sampling efficiency corresponds to the integrated error introduced by sampling with the glass plate, used in Eq. (7) for correction. This is based on a multiplicative error model, where the error (i.e., sampling efficiency) scales linearly with the quantity needing correction (i.e., measured concentration in glass plate samples), without an offset.

## 2.6 Additional measurements and recorded parameters

In experiment B temperature of water and air was measured occasionally with common thermometers. In experiment C, additional parameters were measured for each sample set. Practical salinity and temperature of the water were measured with a Cond 330i with a TetraCon 325 (WTW, Weilheim, Germany) at a depth of about 2 cm below surface. The pH of water was measured at about 1.5 cm below surface with a digital pH meter PH-100 ATC (Voltcraft, Hirschau, Germany), calibrated every morning with buffer solutions with pH of 7.0 and 4.0 (Scharlab, Barcelona, Spain). Air temperature was measured with a common digital thermometer (about 46 cm above water surface) and a liquid expansion thermometer (about 15 cm above water surface, low precision of  $\Delta T = 0.5$  °C). The number of dips, sample volume in mL (or sample weight in gram) and the person who was sampling were recorded in both experiments. In experiment C, the duration of glass plate sampling in seconds was recorded as well.

#### 2.7 Statistical analysis

Data processing, visualization and statistical analysis was performed in Python version 3.11.11 (The Python Language Reference, 2025). The libraries used for data processing and visualization were pandas version 2.3.2 (McKinney, 2010; The Pandas Development Team, 2025), NumPy version 2.3.3 (Harris et al., 2020), seaborn version 0.13.2 (Waskom, 2021) and matplotlib version 3.10.0 (Hunter, 2007; The Matplotlib Development Team, 2024). For statistical analysis, SciPy version 1.16.2 (Virtanen et al., 2020) and statsmodels version 0.14.5 (Seabold and Perktold, 2010) were used. The main function for each statistical calculation is given below. Parameters of the function call are only mentioned if they were changed from default. Observations with missing values were only excluded if the variable concerned was used in the respective statistical analysis. Outliers were included in descriptive statistics and statistical tests, as large variation was expected and removing outliers would potentially remove true values. In boxplots, however, outliers are shown with the common 1.5 times

interquartile range (IQR) in order to present all of the data and to highlight the skewedness. Pair-wise differences between means of treatments and experiments for each trace gas were assessed with Welch's t-test (SciPy, ttest\_ind(...,equal\_var=False)), as for a few cases the heteroscedasticity could not be safely assumed (minimum and maximum variances were off by a factor of more than 3). No multiple testing correction was applied. One-way ANOVA was used to determine if purging of surfactants samples had an effect on the measurements and to identify (categorical) factors (without order) driving well-mixed ratios in Fig. Figure 3 (SciPy, f\_oneway()). The null hypothesis was rejected at p 

## 3.1 Experimental conditions and parameters

## 3.1.1 Water temperature and salinity

in both treatments in experiment B. The temperature in the FW treatment was only measured once at the end of the day (16:30 UTC, 19 April), yielding 20.9 °C. Having been filled the previous day and stored under the SURF roof overnight, the water likely warmed over the course of the day, reaching 20.9 °C at the time of measurement. The AS treatment on the next day started off warmer and cooled down over the course of the day, averaging to 21.5 °C ( $\Delta T = 4.6$  °C). In experiment C (located in lab), a larger range of water temperatures was targeted by using refrigerated water that would heat up in the course of the day to room temperature. This was only partially successful, as the targeted water temperature of <10 °C to start with was not achieved. The FW treatment averaged at 17.6 °C, AS at 19.5 °C and the treatment with artificial surfactants at 20.6 °C (with  $\Delta T$  of 1.2, 3.0 and 1.5 °C respectively). Water temperature of FW treatment overlaps partially with the AS treatment, and the AS treatment partially with the treatment including surfactants, allowing for continuity across treatments. Salinity was set to either zero or to oceanic levels.  $S_P$  was between 28.9 and 34.6 in all treatments with salt amendments. Salinity was kept constant for each treatment level (FW, AS, and each of the SA levels), with exception of the AS treatment without surfactants, which combines two lab days. On 8 August the targeted practical salinity of 34.5 was not reached, but 29.0  $\pm$  0.07. This was corrected to 34.6 on 10 August 2023.

The overall water temperature range across all experiments and treatments was 17.2–23.1 °C. Water temperature was similar

## 3.1.2 Surface activity and enrichment of surfactants

Artificial surfactants were added in experiment C to AS only. Since GC–MS samples were reused for voltammetry analysis, it was tested, if the delayed freezing and the purging had an effect on measured SA. Comparison was performed on a specially collected sample set (n = 6) and across all ULW samples with voltammetry measurements (n = 36). Neither purging (one-way ANOVA failed to reject null hypothesis,  $p_{all} = 0.8864$  with n = 36,  $p_{subset} = 0.5138$  with n = 6) nor the delayed freezing (one-way ANOVA failed to reject null hypothesis,  $p_{all} = 0.3788$  with n = 30) had a significant effect. SA of purged and non-purged samples is thus treated the same (data points shown in Appendix C).

The goal was for the surfactants to reach EF = 1 after wet surfactant TX-100 was added to AS. We targeted a range of SA, so on each day a different amount was added to the tank, increasing over time (Table 2). Mean  $SA_{ULW}$  ranged from 0.160 to 1.515 mg L<sup>-1</sup> TX-100 equiv., while mean  $SA_{SML}$  was much higher between 2.144 to 8.244 mg L<sup>-1</sup> TX-100 equiv., consequently the SML was enriched on average EF from 3.7–11.7 (with three exceptions where EF < 1.0, once for each level of TX-100 added).  $SA_{ULW}$  was usually pretty consistent for one added amount of TX-100 (standard deviation < 0.241 mg L<sup>-1</sup> TX-100 equiv.), whereas  $SA_{SML}$  varied much more with standard deviations from 1.226 to 3.303 mg L<sup>-1</sup> TX-100 equiv., which in turn cause high standard deviation in the EF as well.

Table 2 Surface activity (SA) measurements and corresponding enrichment factors (EF) in experiment C per day in the laboratory in treatment artificial seawater (AS) with surfactants. Each day a different amount of TX-100 was added, increasing over time. For n=2 minimum and maximum values are depicted instead of mean and standard deviation.

|              |                             |                   |          |    | SA                   |          |    |                   |          |  |
|--------------|-----------------------------|-------------------|----------|----|----------------------|----------|----|-------------------|----------|--|
|              | ULW                         |                   |          |    | SML                  |          |    | Enrichment factor |          |  |
| lab day      | [mg $L^{-1}$ TX-100 equiv.] |                   |          |    | $[mg L^{-1} TX-100]$ | equiv.]  |    |                   |          |  |
| 2023         | n                           | mean $\pm SD$     | (median) | n  | mean $\pm SD$        | (median) | n  | mean $\pm SD$     | (median) |  |
| 10<br>August | 2                           | 0.136; 0.185      | (0.160)  | 4  | $1.876 \pm 1.226$    | (2.144)  | 4  | $11.7 \pm 7.6$    | (13.4)   |  |
| 28<br>August | 4                           | $0.762 \pm 0.099$ | (0.774)  | 6  | $2.785 \pm 1.139$    | (3.211)  | 6  | $3.7 \pm 1.5$     | (4.2)    |  |
| 31<br>August | 3                           | $1.442 \pm 0.241$ | (1.515)  | 5  | $6.776 \pm 3.303$    | (8.244)  | 5  | $4.7 \pm 2.3$     | (5.7)    |  |
| all          | 9                           | $0.855 \pm 0.522$ | (0.778)  | 15 | $3.873 \pm 2.926$    | (3.187)  | 15 | $6.1 \pm 5.2$     | (5.0)    |  |

# 315 3.1.3 Sampling duration, sample volume and number of dips

The duration of sampling was recorded for SML samples from right before the first dip until the vial was closed with the rubber stopper as a potential driver for sampling efficiency variability. Sampling duration ranged from 54–235 s over all experiments. Only a few durations were recorded in experiment B in the AS treatment, amounting to 135 s on average. In experiment C, the shortest sampling duration was recorded for the AS treatment with a mean of 112 s. FW sampling took longer with 145 s on average. While the *SA* was increased each day, the sampling duration decreased from 183 s initially (10 August), to 139 s on the second day (28 August) and 129 s on the last day (31 August). Sampling duration greatly depends on the number of dips, which in turn depends on the targeted sample volume. Sampling the treatment with surfactants overall took longer to sample than the AS treatment, because 15 mL were targeted instead of 10 mL for AS.

ULW samples were taken with a pipette, resulting in exact sample volumes of 10 mL and 15 mL respectively. SML samples, on the other hand, showed variation depending on how much water would stick to the glass plate per dip. FW and AS samples were mostly slightly larger than the targeted 10 mL ( $10.8 \pm 1.0$ , n = 30). The samples with the lowest TX-100 added were mostly slightly less than the targeted 15 mL ( $14.3 \pm 0.7$ , n = 4). For the next increase of TX-100, sample volumes were mostly slightly more than the targeted 15 mL ( $15.9 \pm 1.0$ , n = 6). For the highest concentration of TX-100, all samples were below the targeted 15 mL ( $12.5 \pm 1.3$ , n = 6). This is linked to how much sample volume can be gathered per dip (Fig. Figure 1, Fig. Figure 2). In the treatment with the highest TX-100 concentration, another (full) dip would have resulted in too much volume for GC–MS analysis (i.e., not enough headspace). Note, that on 28 August (medium concentration of TX-100) also

half dips were captured (n = 2), i.e., only the wipe of the first side of the glass plate was filled into the vial. The second side was discarded.

The number of dips was recorded to calculate the operational SML thickness (Eq. (1)) and as a potential, additional source of sampling efficiency variability besides sampling duration. To fill about 10 mL of sample without artificial surfactants, three to five dips were required, whereas two to four dips were required to fill about 15 mL with artificial surfactants added (Fig. Figure 1). FW and AS are similar, also across experiments (FW and AS in experiment B 2.4 and 2.9 mL per dip, in experiment C 3.3 and 3.2 mL per dip respectively). In the treatment with artificial surfactants significantly more volume (5.1 mL per dip) was collected per dip, on average, while the volume collected per dip increases with the amount of TX-100 present in the SML (measured as SA), as shown in Fig. Figure 2.

Figure 1 Collected sample volume  $V_{sample}$  for SML samples per dip of the glass plate in the three treatments. Black lines indicate standard deviation.

Figure 2 Sample volume collected per dip in experiment C with the glass plate increases with SA.

An increased sample volume per dip indicates that a thicker layer was sampled (Eq. (1)). Layer thickness ranged from 25–63  $\mu$ m. In experiment B, FW and AS differ at 40  $\pm$  7  $\mu$ m and 48  $\pm$  4  $\mu$ m, while the sampled layer was thinner and more similar in FW and AS in experiment C, with 33  $\pm$  4  $\mu$ m and 32  $\pm$  3  $\mu$ m respectively. In the treatment with artificial surfactants, the thickness increases to an average of 51  $\pm$  10  $\mu$ m.

#### 3.2 Mixing in the tank

Our experiments are based on the premise that SML and ULW have the same trace gas concentration, achieved by mixing the tank before each sampling. In order to test our assumption, a reference ULW sample ( $PA_{ref}$ ) was taken and compared to samples from other depths and locations in the tank (Fig. Figure 3). PA was either taken at reference depth, but at a different location, or at the surface. Almost all ratios ( $PA/PA_{ref}$ ) lie within the range of uncertainty of the measurements (grey shaded area) for all three trace gases. Four sample sets (indicated as grey points) were removed from further calculations (i.e., sampling efficiency), as the additional QC did not indicate poor quality, nor were there deviations from the general sampling procedure noted in the protocols. Four points for isoprene-d5 and three for  $^{13}CS_2$  from 20 April and 08 August were outside the uncertainty range, but were not removed from the data set. Mean values of the ratios (excluding grey points) are  $1.00 \pm 0.05$ ,  $1.01 \pm 0.10$ , and  $1.01 \pm 0.08$  for DMS-d3, isoprene-d5, and  $^{13}CS_2$  respectively. Standard deviations vary between trace gases, with the largest variation for isoprene-d5. Ratios are independent of sampling location, depths, day or experiment (one-way ANOVA failed to reject null hypothesis in all cases).

Figure 3 Test for uniform trace gas concentration in the tanks (well-mixed). The ratios  $PA/PA_{ref}$  (n=37) are shown against date for DMS-d3 (left), isoprene-d5 (middle) and  $^{13}\text{CS}_2$  (right). Black horizontal line depicts  $PA = PA_{ref}$  (i.e., well-mixed). Grey shaded area indicates  $\pm 14.1$  % uncertainty, calculated by error propagation. Grey points (n=4) were identified as not well-mixed. Mean and standard deviation for  $PA/PA_{ref}$  is indicated in upper right of each subplot (excluding grey points). April samplings belong to experiment B, all other samplings are experiment C. All three treatments (FW, AS, AS with surfactants) are included.

380

385

#### 3.3 Measurements of trace gases

Figure 4 Correlation of SML and ULW *PA* for DMS-d3 (left), isoprene-d5 (middle) and <sup>13</sup>CS<sub>2</sub> (right), grouped by treatments fresh water (FW) and artificial seawater (AS) without and with surfactants. Linear fits for <sup>13</sup>CS<sub>2</sub> were performed on the non-logged data, log-log scale is only used to better visualize the very low values.

Figure 3 shows measured PA from treatments FW, AS and AS with surfactants, plotted as SML against ULW, for DMS-d3, isoprene-d5, and <sup>13</sup>CS<sub>2</sub>. SML is always much lower than ULW. The order of magnitude for PA varies between trace gases due to different amounts of trace gas in the spike. DMS-d3 is in the order of 10<sup>4</sup>, isoprene-d5 at 10<sup>3</sup> and <sup>13</sup>CS<sub>2</sub> up to 10<sup>5</sup>. <sup>13</sup>CS<sub>2</sub> shows two distinct clusters of points, as the two spikes used were significantly different in the amount of  ${}^{13}CS_2$ , whereas the amounts for DMS-d3 and isoprene-d5 were similar. For all three gases, a clear linear, positive correlation is visible between SML and ULW. R<sup>2</sup> of the linear fit (SciPy, linregress()) for DMS-d3 and isoprene-d5 are both 0.77 for FW, 0.79 and 0.75 for AS, and much lower at 0.55 and 0.50 for AS with surfactants respectively.  $R^2$  for  $^{13}$ CS<sub>2</sub> is 0.93 for F and 0.95 for AS, and much lower than for the other trace gases for AS with surfactants at 0.37. Note, that the latter data points all fall exclusively into one cluster only, indicating a bad fit without the influence of the large spread present for FW and AS. These values, together with slope and intercept as well as mean PA with standard deviation are reported in Table 3. Mean values and standard deviation visualize the order of magnitude, as within each treatment the amount of spike added differed slightly, causing high standard deviation. This is especially visible for <sup>13</sup>CS<sub>2</sub>, where standard deviation is close to the mean value for FW and AS (includes both clusters), but standard deviation is much lower than the mean for AS with surfactants (only lower value cluster). Slopes differ slightly between gases and treatments. Slopes for DMS-d3 are 0.17 (FW), 0.21 (AS) and 0.14 (AS with surfactants). Slopes for isoprene-d5 are similar, with 0.17 (FW), 0.20 (AS) and 0.12 (AS with surfactants). Slopes for <sup>13</sup>CS<sub>2</sub> are lower, with 0.09 (FW and AS) and 0.10 (AS with surfactants). Intercepts are not zero, but all are much smaller than the respective mean

 $PA_{ULW}$  and therefore considered negligible. Correlation of all data points without distinguishing treatments yields  $R^2$  of 0.80 (DMS-d3), 0.76 (isoprene-d5) and 0.96 ( $^{13}$ CS<sub>2</sub>).

Table 3 Linear regression of PA of SML and ULW samples for DMS-d3, isoprene-d5, and  $^{13}CS_2$ . All  $R^2$  show a strong fit, except when marked with  $^*$  (moderate fit).

|                    |                     |          |                   | 1               | PA     |                        |            |  |  |
|--------------------|---------------------|----------|-------------------|-----------------|--------|------------------------|------------|--|--|
|                    |                     |          | ULW               | SML             | Linear | Linear Regression (OLS |            |  |  |
| Trace gas          | Treatment           | n mean ± |                   | $\pm SD$        | slope  | intercept              | $R^2$      |  |  |
|                    | FW                  | 17       | $14606 \pm 5240$  | 1817 ± 1019     | 0.17   | -679                   | 0.77       |  |  |
| DMC 12             | AS                  | 13       | $26547 \pm 9157$  | $3755 \pm 2153$ | 0.21   | -1796                  | 0.79       |  |  |
| DMS-d3             | AS with surfactants | 16       | $21559 \pm 4685$  | $2805 \pm 873$  | 0.14   | -175                   | $0.55^{*}$ |  |  |
|                    | all                 | 46       | $20399 \pm 7970$  | $2708 \pm 1575$ | 0.18   | -891                   | 0.80       |  |  |
|                    |                     |          |                   |                 |        |                        |            |  |  |
|                    | FW                  | 16       | $719 \pm 282$     | $91 \pm 54$     | 0.17   | -29                    | 0.77       |  |  |
| : 15               | AS                  | 13       | $1334 \pm 475$    | $178 \pm 109$   | 0.20   | -88                    | 0.75       |  |  |
| isoprene-d5        | AS with surfactants | 16       | $1183 \pm 238$    | $150 \pm 42$    | 0.12   | 4                      | $0.50^{*}$ |  |  |
|                    | all                 | 45       | $1062 \pm 422$    | $137 \pm 78$    | 0.16   | -34                    | 0.76       |  |  |
|                    |                     |          |                   |                 |        |                        |            |  |  |
|                    | FW                  | 17       | $56749 \pm 44288$ | $5727 \pm 4236$ | 0.09   | 500                    | 0.93       |  |  |
| 13 cm              | AS                  | 13       | $81274 \pm 90906$ | $7326 \pm 8266$ | 0.09   | 114                    | 0.95       |  |  |
| $^{13}\text{CS}_2$ | AS with surfactants | 16       | $7159 \pm 1385$   | $865 \pm 235$   | 0.10   | 128                    | $0.37^{*}$ |  |  |
|                    | all                 | 46       | $46431 \pm 61976$ | $4488 \pm 5674$ | 0.09   | 331                    | 0.96       |  |  |

#### 3.4 Sampling efficiency for fresh water and artificial seawater

The sampling efficiency of the trace gases is calculated using the ratio of measured PA in SML over measured PA in ULW 395 (Eq. (5)). Due to mixing in the tank the theoretical maximum value for the sampling efficiency is, thus, 1.0 (or 100 %), i.e., when  $PA_{SML} = PA_{ULW}$ .

Figure 5 shows sampling efficiency for the studied trace gases in FW and AS treatment, separated by experiment. Overall the three trace gases are similar, but there are significant differences between the experiments (fails to reject null hypothesis with p 

Figure 5 Sampling efficiency for DMS-d3 (left), isoprene-d5 (middle) and  $^{13}$ CS<sub>2</sub> (right), separated by experiment, coloured by treatments fresh water (FW, blue) and artificial seawater (AS, orange). Box width shows 25 to 75 percentile, centre line denotes the median, diamonds denote the mean, open circles denote outliers, and whiskers show minimum and maximum. The number of data points in experiment B is  $n_{FW} = 11$  ( $n_{FW} = 10$  for isoprene-d5) and  $n_{AS} = 6$ , in experiment C  $n_{FW} = 6$  and  $n_{AS} = 7$ .

Mean sampling efficiency for DMS-d3 in experiment B is  $0.105 \pm 0.018$  (n = 11) in FW, decreasing to  $0.090 \pm 0.020$  (n = 6) in AS, though the difference is not significant (p = 0.15, outliers in Fig. Figure 5 included). In experiment C, the mean sampling efficiency for DMS-d3 in both treatments is higher than in experiment B, with  $0.144 \pm 0.034$  (n = 6) in FW increasing to  $0.164 \pm 0.033$  (n = 7) in AS. The difference between FW and AS within experiment C is not significant (p = 0.32). This overall picture repeats for isoprene-d5 and  $^{13}$ CS<sub>2</sub>. In experiment B, mean sampling efficiency for isoprene-d5 is  $0.112 \pm 0.025$  (n = 10) in FW, reducing to  $0.079 \pm 0.018$  (n = 6) in AS, whereas in experiment C sampling efficiency is overall higher, with  $0.143 \pm 0.035$  (n = 6) in FW and then further increasing to  $0.158 \pm 0.036$  (n = 7) in AS. For  $^{13}$ CS<sub>2</sub>, sampling efficiency in experiment B averages to  $0.101 \pm 0.025$  (n = 11) in FW and lowers to  $0.079 \pm 0.019$  (n = 6) in AS, whereas it is overall higher in experiment C with  $0.131 \pm 0.032$  (n = 6) in FW and rising to  $0.158 \pm 0.031$  (n = 7) in AS. Similar to DMS-d3, the differences within both experiments between treatments are not significant (Table 5), except for isoprene-d5 in experiment B (p = 0.009). Standard deviations of sampling efficiency are slightly larger in experiment C than in B for all three trace gases and differ significantly between treatments FW and AS (Levene's test for equal variances failed to reject null hypothesis with p = 0.045, 0.0497 and 0.033 for DMS-d3, isoprene-d5, and 0.045 (0.0497 and 0.033 for DMS-d3, isoprene-d5, and AS for DMS-d3 is 0.0074 and 0.013, for isoprene-d5 is 0.0080 and 0.014 and for 0.045 and 0.0073 and 0.013. SE overall (n = 30) are 0.0070, 0.0074 and 0.007 for DMS-d3, isoprene-d5, and 0.014 and for 0.0075 and 0.0073 and 0.013. SE overall (n = 30) are 0.0070, 0.0074 and 0.0075 for DMS-d3, isoprene-d5, and 0.0075 and 0.0073 and

There is no significant difference between FW in experiment B and C for any of the three trace gases (except for DMS-d3, p = 0.038), but in AS the differences are significant for all three trace gases (p < 0.001), see Table 5. The medium used in FW was ultrapure water, from different devices, but same models. The media used for AS, on the other hand, were different in the composition of the salts and other substances added, though also  $S_P$  slightly varied.

Table 4 Summary statistics of sampling efficiency for DMS-d3, isoprene-d5, and  $^{13}$ CS<sub>2</sub> in experiment B and C in fresh water (FW) and artificial seawater (AS) without and with surfactants treatment, as visualized in Fig. Figure 5 and Fig. Figure 6. † indicates in which group one data point had to be removed for isoprene-d5 ( $PA_{SML}$  < LOD).

|            | Summary Statistics  |                 |                   |          |  |                   |          |                   |                    |  |
|------------|---------------------|-----------------|-------------------|----------|--|-------------------|----------|-------------------|--------------------|--|
|            |                     |                 | DMS-d3            |          |  | isoprene          | e-d5     | <sup>13</sup> CS  | $^{13}\text{CS}_2$ |  |
| Experiment | Treatment           | n               | mean $\pm SD$     | (median) |  | mean $\pm SD$     | (median) | mean $\pm SD$     | (median)           |  |
| В          | FW                  | 11 <sup>†</sup> | $0.105 \pm 0.018$ | (0.108)  |  | $0.111 \pm 0.025$ | (0.111)  | $0.101 \pm 0.025$ | (0.102)            |  |
| D          | AS                  | 6               | $0.090 \pm 0.020$ | (0.088)  |  | $0.079 \pm 0.018$ | (0.082)  | $0.079 \pm 0.019$ | (0.079)            |  |
|            |                     |                 |                   |          |  |                   |          |                   |                    |  |
|            | FW                  | 6               | $0.144 \pm 0.034$ | (0.135)  |  | $0.143\pm0.035$   | (0.133)  | $0.131 \pm 0.032$ | (0.123)            |  |
| C          | AS                  | 7               | $0.164 \pm 0.033$ | (0.169)  |  | $0.158\pm0.036$   | (0.163)  | $0.158 \pm 0.031$ | (0.160)            |  |
|            | AS with surfactants | 16              | $0.130 \pm 0.025$ | (0.131)  |  | $0.127 \pm 0.024$ | (0.124)  | $0.121 \pm 0.024$ | (0.120)            |  |

Sampling efficiency as a function of water temperature, salinity, spike volume per litre added (i.e., proportional to trace gas concentration) and number of dips was investigated using MLR models (n = 19 complete observations of 30). Note, that 10 of 11 data points in experiment B treatment FW were excluded, because of missing temperature measurements, and one of seven in experiment C treatment AS, because of unrecorded number of dips. The MLR models were significant for all three trace gases (F(4,14) > 5.6, p < 0.01) and explain about 60 % of the variance observed in sampling efficiency (adjusted  $R^2$  is 0.63, 0.60 and 0.65 for DMS-d3, isoprene-d5, and  $^{13}$ CS<sub>2</sub> respectively). The models predicted average ( $\pm$  SE) sampling efficiency of 0.129  $\pm$  0.007, 0.127  $\pm$  0.008 and 0.120  $\pm$  0.007 for DMS-d3, isoprene-d5, and  $^{13}$ CS<sub>2</sub>, when all predictors were at their mean values (p 

Table 5 Test statistics of sampling efficiency for DMS-d3, isoprene-d5, and 13CS2 in experiment B and C in fresh water (FW) and artificial seawater (AS) treatment, as visualized in Fig. Figure 5.  $^{\dagger}$  indicates in which group one data point had to be removed for isoprene-d5 ( $PA_{SML} < LOD$ ).  $^*$ ,  $^{**}$  and  $^{***}$  mark where the null hypothesis was rejected, with p-values <0.05, <0.01 and <0.001 respectively.

|     |        |                 | Welch's t-t     | test            |                      |  |
|-----|--------|-----------------|-----------------|-----------------|----------------------|--|
|     |        |                 | DMS-d3          | isoprene-d5     | $^{13}\mathrm{CS}_2$ |  |
| Gro | oups   | n               | <i>p</i> -value | <i>p</i> -value | <i>p</i> -value      |  |
| I   | B<br>C |                 | <0.001***       | <0.001***       | <0.001***            |  |
| (   |        |                 | <0.001          | <0.001          | <0.001               |  |
| F   | W      | 13 <sup>†</sup> | 0.40            | 0.00            |                      |  |
| A   | .S     | 17              | 0.48            | 0.92            | 0.52                 |  |
|     | FW     | $6^{\dagger}$   | 2.45            | 0.000**         | 0.063                |  |
| В   | AS     | 11              | 0.15            | 0.009**         |                      |  |
| G   | FW     | 7               | 0.22            | 0.15            | 0.45                 |  |
| С   | AS     | 6               | 0.32            | 0.46            | 0.15                 |  |
|     | В      | $6^{\dagger}$   | 0.020*          |                 |                      |  |
| FW  | C      | 11              | 0.038*          | 0.090           | 0.083                |  |
|     | В      | 7               |                 |                 |                      |  |
| AS  | C      | 6               | <0.001***       | <0.001***       | <0.001***            |  |

#### 450 3.5 Sampling efficiency with artificial surfactants

To mimic in-situ conditions more closely, the experiments were extended to include a treatment of AS with surfactants.

When artificial surfactants (TX-100) were added to AS (Fig. Figure 6), the mean sampling efficiency for all of the three trace gases reduced even below the FW treatment, DMS-d3 to 0.130, isoprene-d5 to 0.127 and <sup>13</sup>CS<sub>2</sub> to 0.121 (Table 4). Standard deviations are also smaller than in FW and AS treatment (0.025 for DMS-d3, 0.024 for isoprene-d5 and <sup>13</sup>CS<sub>2</sub>). The range (difference from minimum to maximum) is still similar to the range observed in FW and AS for each of the three trace gases.

Sampling efficiency decreases with increasing  $SA_{SML}$  (Fig. Figure 7). It decreases fast from SA = 0 until about SA = 3.0. Above SA = 3.0 the values seem to arrange around a constant value of sampling efficiency. Linear fit is weak ( $R^2$  between 0.26 and 0.28 for all three trace gases). The slope is negative and small with about -0.005 per mg L<sup>-1</sup> TX-100 equiv. and intercepts are between 0.146 to 0.154 for the three trace gases (

Table 6). Though small, the slope is significantly different from no slope (p < 0.05 for all three trace gases). An exponential fit was only possible for DMS-d3 and  $^{13}$ CS<sub>2</sub>, for which it explains slightly more variability than the linear fit ( $R^2$  between 0.42 and 0.44). When treatment FW was included exponential fit was only possible for DMS-d3 (not shown). The exponential fit approaches 0.12 and 0.11 for DMS-d3 and  $^{13}$ CS<sub>2</sub> with a small amplitude of 0.05 for both at a decay rate of -0.68 and -0.88. The exponential fit is added for visualization in Fig. Figure 7.

Figure 6 Sampling efficiency for DMS-d3 (left), isoprene-d5 (middle) and  $^{13}$ CS<sub>2</sub> (right) when artificial surfactants are added to artificial seawater (AS with surfactants) from experiment C ( $S_P = 31.1 \pm 2.9$ ). Fresh water (FW) and artificial seawater without surfactants (AS) are depicted as well for reference. Box width shows 25 to 75 percentile, centre line denotes the median, diamonds denote the mean, open circles denote outliers, and whiskers show minimum and maximum. The number of data points is  $n_{FW} = 6$ ,  $n_{AS} = 7$  and  $n_{SA} = 16$ .

Table 6 Curve fitting for sampling efficiency against SA<sub>SML</sub> in experiment C, treatments with AS only.

|                 |                          | Linear Fit         |       | Exponential Fit |                                     |   |
|-----------------|--------------------------|--------------------|-------|-----------------|-------------------------------------|---|
| Trace gas       | $n$ slope $\pm SE$       | intercept $\pm SE$ | $R^2$ | <i>p</i> -value | fitted function $R^2$               | ? |
| DMS-d3          | $22 \ -0.0054 \pm 0.002$ | $0.154 \pm 0.008$  | 0.26  | 0.015*          | $0.048e^{-0.68x} + 0.12 \qquad 0.4$ | 2 |
| isoprene-d5     | $22 \ -0.0055 \pm 0.002$ | $0.151 \pm 0.008$  | 0.28  | 0.012*          | fit not possible                    |   |
| $^{13}$ CS $_2$ | $22 \ -0.0054 \pm 0.002$ | $0.146 \pm 0.008$  | 0.27  | 0.013*          | $0.047e^{-0.88x} + 0.11$ 0.4        | 4 |

Figure 7 Sampling efficiency for DMS-d3 (left), isoprene-d5 (middle) and <sup>13</sup>CS<sub>2</sub> (right) against surface activity (*SA*) in experiment C. Exponential fit is added as aid for visualization.

Sampling efficiency as response to water temperature, salinity, spike volume per litre added (i.e., proportional to trace gas concentration), number of dips and SA in the SML was investigated using MLR models for each trace gas (n = 27 complete observations of 29). The MLR models were not significant for any of the three trace gases (F(5,21) > 1.9, p > 0.09) and explained less than 25 % of the variance observed in sampling efficiency (adjusted  $R^2$  is 0.25, 0.22 and 0.23 for DMS-d3, isoprene-d5, and  $^{13}$ CS<sub>2</sub> respectively). The model predicted average ( $\pm SE$ ) sampling efficiency of 0.139  $\pm$  0.006, 0.137  $\pm$  0.006 and 0.131  $\pm$  0.006 for DMS-d3, isoprene-d5, and  $^{13}$ CS<sub>2</sub>, when all predictors were at their mean values (p < 0.001). Spike volume per litre is a significant, positive predictor for sampling efficiency for only DMS-d3 anymore (p = 0.029,  $\beta = 0.0213$ ,  $s_{spike} = 0.94 \,\mu$ L L<sup>-1</sup>). Instead, water temperature is now a significant, still negative predictor for all three trace gases (p < 0.05,  $\beta$  is -0.0324, -0.0309 and -0.0321 for DMS-d3, isoprene-d5, and  $^{13}$ CS<sub>2</sub>,  $s_{Tw} = 1.4$  °C). SA in the SML was a significant, negative predictor for  $^{13}$ CS<sub>2</sub> only (p = 0.041,  $\beta = -0.0137$ ,  $s_{SA,SML} = 2.91$  mg L<sup>-1</sup> TX-100 equiv.), though DMS-d3 is on the limit (p = 0.05), while isoprene-d5 is non-significant (p = 0.07). Neither number of dips (p > 0.8), nor salinity (p is 0.247 and 0.206 for DMS-d3 and isoprene-d5, but much lower at 0.066 for  $^{13}$ CS<sub>2</sub>) were a significant predictor for any of the trace gases. See Appendix E Results linear and multiple linear regressions with artificial surfactants from Sect. 3.5 for details of MLR, linear regression results and additional plots.

#### 490 4 Discussion

#### 4.1 Experimental setup

Water temperature in this study covered a range of  $\Delta T = 5.9$  °C. Sea surface temperature in the field encompasses a wider range. At Boknis Eck station, for example, the range studied here is only observed during the summer season (Laß et al., 2013).

In our treatments, intermediate salinity was excluded, focussing on fresh water and seawater. We do not expect to see any improvement in our results by including intermediate salinity, as salinity was a non-significant predictor for sampling efficiency. However, the choice of salt (Tropic Marin® Pro-Reef Sea Salt vs ordinary aquarium salt) may have had an effect, which is attributed to either the composition of the salt (affecting the solubility, e.g., (Weisenberger and Schumpe, 1996)), other substances or the alkalinity of the water, rather than the salinity. The increase in targeted sample volume from 10 mL (FW, AS) to 15 mL (AS with surfactants) did not affect  $PA_{ULW}$ . However, we could not assess whether  $PA_{SML}$  differed between 10 and 15 mL. The increase in volume is achieved by dipping the glass plate more often, potentially affecting the sampling efficiency. Since the number of dips was a non-significant predictor for sampling efficiency in the MLR, the effect on  $PA_{SML}$  and consequently on sampling efficiency, is considered negligible. Finally, a range of saturation states was not investigated. We only performed experiments with oversaturated conditions in the water, but we expect that the concentration gradient (magnitude and direction) has an effect on sampling efficiency.

#### 4.1.1 Mixing in the tank

Figure 3 shows that the tank was well-mixed on all days, because almost all the ratios  $PA/PA_{ref}$  are within the acceptable uncertainty range. The four points from 20 April and 08 August, which were not acceptable for isoprene-d5, and the three for  $^{13}\text{CS}_2$ , are still considered as well-mixed, since the ratios are acceptable for DMS-d3. The mean values of the ratios (excluding grey points) are all close to 1.0 (i.e.,  $PA = PA_{ref}$ ), supporting that the tank was well-mixed without any physically driven gradients present or forming, as then the mean would likely show an offset. This is further supported by the ratios with surface samples on 20 April, sampled as close as possible to the surface. Due to the oversaturation in the tank and near-zero atmosphere, we hypothesize that  $PA_{surface} 

530

540

545

550

555

develop is effectively shorter, since the turbulent motion from mixing took about 1 min to subside to a level when diffusion starts to dominate. This also causes the DBL thickness to stay thinner. Additionally, the geometry of the tank limits the maximum thickness of the DBL, whereas in our calculation we assume a monotonic increase of the thickness over time. For our setup a thickness of 0.5 mm is more reasonable. This is still much larger than the thickness that was sampled by one dip of the glass plate, though with the limited tank surface it might be more appropriate to compare the DBL thickness with the sample volume distributed over the full tank surface, which yields that about 0.1 mm of the surface that is sampled away by one SML sample. Note, that the 0.79 reflects the concentration in the DBL at the instant of time right before the first dip of the glass plate. Our calculations are not valid anymore when the sampling disturbs the tank. The box model used is explained in detail in Appendix F.

The estimate of the concentration including diffusive losses is much higher than our sampling efficiency, which indicates that the DBL dilution is not dominating our sampling efficiency estimates. This is especially true, since the diffusive losses are overestimated under the assumptions we made in our calculations.

#### 4.1.3 Influence of surfactants addition

The decrease of sampling efficiency with increasing TX-100 concentration (Fig. Figure 7) in our lab experiments is unexpected. It is hypothesized that air-sea gas exchange of trace gases decreases with increasing  $SA_{SML}$ , resulting from (1) the wave dampening effect of surfactants that in turn reduces the turbulence, and, thus, the turbulence driven gas exchange, and (2) the surfactants enriched layer acting as a physicochemical barrier (Garbe et al., 2014, 2.2.7 Surface Films). The decrease in gas exchange is a net effect of these two factors and depends on SA. Although there was no wind on our lab experiments, some turbulence was introduced by mixing and sampling, leading to the hypothesis that sampling efficiency would increase with SA. It should be noted within this discussion that natural slicks have been related to exceeding a threshold of 1 mg  $L^{-1}$ TX-100 equiv. up to and beyond 3 mg  $L^{-1}$  TX-100 equiv. (Barthelmeß and Engel, 2022). 12 SML samples were >1 mg  $L^{-1}$ TX-100 equiv. and 9 were >3 mg L<sup>-1</sup> TX-100 equiv. (of 15), indicating that we reached beyond the natural range of SA in the SML. Natural surfactant pools consist of a mixture of substances, which vary greatly depending on the biological and chemical processes on-going in the SML and in their physicochemical effect (Engel et al., 2017). Though TX-100 as a wet surfactant is considered a good choice for a model surfactant, it is not ideal. Adenaya et al. (2021), for example, show that DBL thickness with TX-100 is up to 30 % thicker than with natural surfactants for  $SA = 2 \text{ mg L}^{-1} \text{ TX-100}$  equiv., therefore, diluting the concentration on the glass plate. Furthermore, we could not rule out a change in the trace gas solubility or diffusivity in surfactants instead of (sea)water. Insufficient purge time caused by a potential decrease in purge efficiency with increasing SA (i.e., mostly affecting  $PA_{SML}$ ) was ruled out as reason for the decrease in sampling efficiency. Also micelle formation was ruled out, as  $SA_{SML}$  was much lower than critical micelle concentration (Mukerjee and Mysels, 1971).

We did observe that surfactants were not well-mixed (EF on average >1.0, Table 2). At about SA = 1.5 mg L<sup>-1</sup> TX-100 equiv. the surface of the water body is covered 100 % with TX-100, beyond which the coverage cannot increase more. Any excess TX-100 mixes further into the ULW, increasing the SML thickness (personal communication Falko Asmussen-Schäfer). EF

560

575

580

585

590

of surfactants for the same level of TX-100 added in our experiments varied strongly, including EF 

595

600

605

610

reactive trace gases, like N<sub>2</sub>O. A small, test data set (data not shown) of N<sub>2</sub>O glass plate and ULW samples (12 SML–ULW pairs) that we collected using artificial seawater ( $S_P = 30.2$ ,  $T_{water} = 22.9 \pm 0.7$  °C) in a tank (equilibrated with lab air over more than 24 h), showed mean sampling efficiency of 0.97  $\pm$  0.12, indicating that the  $\Delta C$  and the corresponding equilibration behaviour of the trace gas play a major role. Unfortunately, this effect was not covered in this study, as the three chosen trace gases were too similar to capture this difference.

#### 4.3 What drives sampling efficiency?

Sampling efficiency in general is driven by turbulence effects, the equilibration between water on the glass plate and the surrounding air, potential adsorption effects (Walker et al., 2016) and photochemical processes (Zemmelink et al., 2005). Note, that chemical and biological reactions are excluded from this study and not considered to be factors for driving sampling efficiency here. The dilution of SML samples by ULW water due to varying sampling thicknesses, is also excluded from our concept of sampling efficiency, because  $C_{SML} = C_{ULW}$ . Turbulence effects and equilibration can both reduce and increase sampling efficiency, depending on the concentration gradient from the glass plate towards the surrounding air. Photochemical processes heavily depend on the trace gas, light intensity and duration of exposure. Sampling efficiency will decrease with degradation and increase with production processes, however, the increase would be artificial (i.e., not due to sampling effects). All experiments were performed indoors and UV light, therefore, was filtered out. Adsorption to the glass plate, the wiper and the funnel always reduces sampling efficiency by removing molecules from the sample, though, the effective amount is hard to quantify. Adsorption losses are deemed negligible in our setup, because of the regular cleaning of all used materials. Turbulence increases the effect of equilibration many times and we kept this at a minimum by omitting wind influence. The presence of substances other than trace gases, e.g., surfactants, can have an enhancing, reversing or neutral effect on turbulence and equilibration. In this study the concentration gradients were always going towards the near-zero atmosphere concentration, therefore, we hypothesize that the turbulence introduced by the sample handling during glass plate sampling decreases sampling efficiency. Additionally, the sampling efficiency is expected to further decrease, due to fast paced equilibration along the high concentration gradient (from oversaturated water to near-zero atmosphere) and the short-lived, highly reactive species with low solubility that we use.

The MLR (explains about 60 % variance) identifies spike volume per litre (positive) and water temperature (negative) as significant predictors for DMS-d3 and  $^{13}$ CS<sub>2</sub> sampling efficiency in FW and AS. For isoprene-d5 it is spike volume per litre (positive) and the number of dips (positive). This changes with the addition of surfactants. The MLR (explains less than 25 % of variance). Spike volume per litre (positive) and water temperature (negative) are significant predictors for DMS-d3 sampling efficiency in FW and AS without and with surfactants.  $SA_{SML}$  is almost a significant predictor (negative). The only significant predictor for isoprene-d5 sampling efficiency is water temperature (negative). Water temperature (negative) and  $SA_{SML}$  are significant predictors for  $^{13}$ CS<sub>2</sub> sampling efficiency. It is reasonable that the spike volume per litre is correlated with sampling efficiency, as it is proportional to the concentration of the trace gases in the tank, which in turn is proportional to the  $\Delta C$  (as  $C_{air}$  = constant  $\approx$  0). The identified, negative relationship of sampling efficiency and water temperature matches the

expectations and the findings by. Most likely this effect relates to the negative relationship of solubility and water temperature, highlighting that the solubility is a potential candidate to be used to predict sampling efficiency. Furthermore, water temperature increases the speed of molecular diffusion, which in turn would also further reduce sampling efficiency. Salinity is not significant for any of the trace gases. Although, salinity has an effect on solubility (Weisenberger and Schumpe, 1996), it is much lower than the effect of temperature. There seems to be an effect from the kind of water used though, as the two AS media used (coral reef seawater and ultrapure water with ordinary aquarium salt) differ significantly, e.g., sampling efficiency for DMS-d3 in AS is almost two times higher in experiment C than in experiment B. Within this study, it cannot be determined whether this can be attributed to the salt composition, other substances mainly present in the coral reef seawater or any of the other properties of the coral reef seawater, like the alkalinity. Surprisingly the number of dips was only significant in one case, with a predicted positive relationship, which would imply a larger sampling efficiency with more dips. This is counter to the hypothesis, that the sampling duration (which is a multiple of the number of dips) decreases sampling efficiency, and this result is therefore deemed a mathematical construct. Although each dip has its individual sampling efficiency, it is to be expected that the number of dips affects the overall variability of our sampling efficiency estimate, but not the mean sampling efficiency. However, the number of dips is closely linked to sampling duration. The sampling duration affects how long the vial is left open, potentially resulting in a change of the mean sampling efficiency. However, our lab experience has shown that leaving the vial open has a negligible effect and non-significant results for the number of dips from the MLR supports this. SA<sub>SML</sub> was included in MLR as the factor driving sampling efficiency, though SA<sub>ULW</sub> seemed to have an effect on PA as well. We expected a positive relationship with sampling efficiency, since field studies show a decrease in gas exchange for increased  $SA_{SML}$ , i.e., more retained in SML. Instead, the model and linear regressions (Appendix E) show negative correlations. Most recent gas exchange reduction publications look into wind-driven effects, which is why those are not directly applicable here, but Schmidt and Schneider (2011) show a decrease in gas exchange for stirring samples with surfactants only, highlighting that a positive correlation is still more likely. The observed negative correlation might be related to the differing sampling thickness (less dips required with surfactants, i.e., larger thickness sampled), though with the mixing we should have achieved a homogeneous concentration throughout the tank (Fig. Figure 3). Even if there had been a negative gradient towards the SML (e.g., DBL, due to outgassing to the near-zero atmosphere), the increased sampling thickness would have put proportionally more ULW into the sample, i.e., increasing  $C_{SML}$  and causing an (artificial) increase in sampling efficiency (i.e., positive correlation). However, Adenaya et al. (2021) show that the thickness of the DBL increases with the addition of artificial surfactants (as opposed to natural surfactants), which would counteract the effect of the sampling thickness. The final assessment, though, is difficult, as the increase of the DBL cannot be quantified in our setup. Though  $SA_{SML}$  was much higher than natural values, it was still very far from the critical micelle concentration of TX-100, which is why the negative correlation can also not be explained by micelle formation. It is possible that the surfactants affect the homogeneous mixing of the trace gases. If surfactants decrease the trace gas concentration with increasing SA, this would explain the negative correlation and the decrease in mean sampling efficiency. The trace gases would mix less into the SML than in the ULW (as  $SA_{SML} > SA_{ULW}$ ), decreasing sampling efficiency.

Finally, the data is relatively scattered and the linear correlation is weak, which might be an indication that we are not capturing a single, physical relationship, but a net effect of entangled factors.

Table 7 Sampling efficiency summarized from treatment FW and AS. MLR is multiple linear regression,  $T_{water}$  is the water temperature and  $C_{ULW}$  the concentration of trace gas in the ULW. <sup>(1)</sup>Values copied from solubility compilation by Sander (2023). <sup>(2)</sup>Value calculated for T = 25 °C and  $S_P = 0$  from Saltzman et al. (1993). Diffusivity in (sea)water is not published (n.p.) for isoprene or CS<sub>2</sub>. <sup>(3)</sup>Due to their analogous structures, CO<sub>2</sub> diffusivity can be used as an approximation for CS<sub>2</sub>, i.e.,  $D_{CO2} = 1.88 \times 10^{-9}$  m² s<sup>-1</sup> at T = 25 °C and  $S_P = 0$  (Mazarei and Sandall, 1980). Global estimates of atmospheric mixing ratios are taken from <sup>(4)</sup>Liss et al. (2014), <sup>(5)</sup>Ferracci et al. (2024), Southern Ocean only, and <sup>(6)</sup>Lennartz et al. (2020).

|           | Samplin                          | g efficiency                               | Solubility <sup>(1)</sup>    | Diffusivity Atmosphe                    | ric |
|-----------|----------------------------------|--------------------------------------------|------------------------------|-----------------------------------------|-----|
|           | Data set                         | MLR Drivers                                | $\times 10^{-3}$             | ×10 <sup>-9</sup> mixing rat            | ios |
| Trace gas | $n$ mean $\pm SE$ min-max        | <i>n</i> intercept $\pm SE$                | $[\bmod \ m^{-3} \ Pa^{-1}]$ | $[m^2 s^{-1}]$ [ppt]                    |     |
| DMS       | $30  0.12 \pm 0.01  0.07 - 0.21$ | 19 $0.13 \pm 0.01$ $T_{water}$ , $C_{ULW}$ | 5.3                          | 1.35 <sup>(2)</sup> ~100 <sup>(4)</sup> |     |
| isoprene  | $29  0.12 \pm 0.01  0.05 0.21$   | 19 $0.13 \pm 0.01$ $C_{ULW}$ , dips        | 0.13                         | n.p. 42.4 <sup>(5)</sup>                |     |
| $CS_2$    | $30  0.12 \pm 0.01  0.05 - 0.20$ | 19 $0.12 \pm 0.01$ $T_{water}$ , $C_{ULW}$ | 0.61                         | n.p. $^{(3)}$ ~50 $^{(6)}$              |     |

#### 5 Conclusion and outlook

In this study we set up experiments to determine sampling efficiency of the glass plate technique for the short-lived trace gases DMS, isoprene and  $CS_2$  in the SML. To understand the drivers of sampling efficiency we varied water temperature, salinity and trace gas concentration. To mimic in-situ conditions, we added three different levels of surfactants in one treatment. Sampling efficiency ( $\pm$  *SE*) is low at about 0.12  $\pm$  0.01, but consistent for all three trace gases, while the drivers only have a low impact in view of the overall uncertainty of the sampling efficiency, though water temperature and trace gas concentration were significant for almost all cases. Note, that surfactants in the SML exceeded natural values by far, and had an unexpected negative correlation with sampling efficiency.

Given the consistency we conclude that meaningful analysis of SML trace gas samples in the field is possible. However, we do not recommend to apply the sampling efficiency quantitatively to correct concentrations of SML samples, given the uncertainty present in our sampling efficiency estimates added to by the uncertainty of factors in the field that have not been attributed in this study (e.g., wind). Instead we propose to use the intercept from the MLR fit with the *SE* of the sampling efficiency as a limit to categorize ratios of measured SML over ULW concentrations for enrichment (Table 7) identifying those above the sampling efficiency as cases of net production. Our results will be applied to samples of DMS, isoprene and CS<sub>2</sub> from a mesocosm study (Bibi et al., 2025) and from two cruises, one in the North Sea (2024) and one in the Baltic Sea (2025), to further investigate the potential to explain SML trace gas processes and link them to air-sea exchange.

Our findings are a first step towards utilizing glass plate sampling for trace gases, yet, they also highlight the need for further investigation. We studied highly reactive, short-lived trace gases here, for which the concentration gradient usually is large. A preliminary test that we performed at presumably equilibrium conditions for the relatively inert, long-lived trace gas  $N_2O$  have

https://doi.org/10.5194/egusphere-2025-5361 Preprint. Discussion started: 17 November 2025

© Author(s) 2025. CC BY 4.0 License.

EGUsphere Preprint repository

shown a contrasting near 100 % sampling efficiency, highlighting the role of the concentration gradient and divergence from equilibrium on sampling efficiency. We, therefore, strongly recommend to extend our results to less reactive and long-lived trace gases, like the prominent greenhouse gases N<sub>2</sub>O and CH<sub>4</sub> to complement our findings. As in field studies of gas exchange wind is an important driver, there is also the need to investigate how sampling efficiency is affected by wind in combination with the trace gas concentration gradient between sample water and air, and whether a threshold of wind speed exists above which glass plate samples do not contain the trace gas anymore. The mesh screen is also widely used. Repeating the study with the mesh screen, could increase the availability of trace gas measurements of the SML. Finally, the relationship between sampling efficiency on the one hand, and water temperature and surfactants on the other hand, needs more in-depth investigation. By unravelling the underlying physical relationships, it might become possible to not only categorize enrichment of trace gases in the SML, but to correct the individual measurements of trace gases in the SML.

#### 695 **Data availability**

All data will be archived and made available to the scientific community at the PANGAEA database upon doi assignment. In the meantime, data are available from the authors upon request.

#### **Author contribution**

Lea Lange: Conceptualization, Data curation, Formal analysis, Investigation, Project administration, Visualization, Writing (original draft preparation), Writing (review and editing)

Dennis Booge: Conceptualization, Supervision, Writing (review and editing)

Hendrik Feil: Investigation, Writing (review and editing)

Josefine Karnatz: Investigation, Writing (review and editing)

Ina Stoltenberg: Supervision, Writing (review and editing)

Hermann W. Bange: Conceptualization, Funding acquisition, Resources, Supervision, Writing (review and editing)

Christa A. Marandino: Conceptualization, Funding acquisition, Resources, Supervision, Writing (review and editing)

## **Competing interests**

HWB is a member of the editorial board of Biogeosciences.

## Acknowledgements

We would like to thank Edgar Cortés, Alisa Wüst, Carola Lehners, Mariana Ribas-Ribas and Oliver Wurl for providing us with support, facilities and consumables during experiment B at ICBM, Carl von Ossietzky Universität Oldenburg. We would

like to thank our student assistant Fenja Möller for supporting us with sampling during experiment C at GEOMAR Helmholtz Centre for Ocean Research Kiel. We would also like to thank our student assistant Samira Linder for helping with surfactants measurements at GEOMAR Helmholtz Centre for Ocean Research Kiel. This is a contribution to the international Surface Ocean-Lower Atmosphere Study (SOLAS), which receives funding and support from the Scientific Committee on Oceanic Research (e.g., NSF Grant OCE-1840868 and 2140395), the iCACGP, the WCRP and Future Earth, and the State Key Laboratory of Marine Environmental Science, China.

#### **Financial support**

This research was supported by the project "Biogeochemical processes and Air-sea exchange in the Sea-Surface microlayer 720 (BASS)", which was funded by the German Research Foundation (DFG) under Grant No 451574234.

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

# **Appendix**

#### A Preliminary test

Experiment A was conducted on three consecutive days between 13–15 March 2023, using one of our incubation baths as tank, filled with ultrapure water. We did these preliminary tests to learn about handling, measurements, mixing in the tank and sampling efficiency calculation from PA. As no glass plate had been available, we used a mesh screen. Targeted sample volume was 60 mL. Figure A1 depicts the ratios of  $PA/PA_{ref}$  in experiment A, analogous to Fig. Figure 3 (experiment B and C). Figure A2 depicts the resulting sampling efficiency.

Figure A1 Test for uniform trace gas concentration in the tank (well-mixed) in experiment A with mesh screen. The ratios  $PA/PA_{ref}$  (n = 16 for DMS-d3 and isoprene-d5, n = 9 for  $^{13}CS_2$ ) are shown against date for DMS-d3 (left), isoprene-d5 (middle) and  $^{13}CS_2$  (right). Black horizontal line depicts  $PA = PA_{ref}$  (i.e., well-mixed). Grey shaded area indicates  $\pm 14.1$  % uncertainty. Mean and standard deviation for  $PA/PA_{ref}$  is indicated in upper right of each subplot (including all shown points).

9 of 16 points lie within the uncertainty range for DMS-d3 and isoprene-d5, and4 of 9 for  $^{13}$ CS<sub>2</sub>. Therefore, it seems the tank was overall less well-mixed than in experiment B and C. However, no points are excluded from calculation of sampling efficiency, as experiment A serves as a test. This results in average ratios ( $\pm$  standard deviation) of PA/PAref of 0.98  $\pm$  0.21, 0.99  $\pm$  0.32, and 1.03  $\pm$  0.29 for DMS-d3, isoprene-d5, and  $^{13}$ CS<sub>2</sub> respectively. Sampling efficiency is 0.30  $\pm$  0.08 (n = 13), 0.27  $\pm$  0.09 (n = 13), and 0.26  $\pm$  0.09 (n = 10) for DMS-d3, isoprene-d5, and  $^{13}$ CS<sub>2</sub> respectively. Mean sampling efficiency and its standard deviation are, thus, larger here than with the glass plate in experiment B and C (compare with FW in Table 4). The preliminary test with the mesh screen was less well constrained, which might explain the larger variation. The mesh screen might be more efficient, since wiping is not required. Also, the mesh screen captures more volume per dip, reducing the overall sampling duration. Furthermore, the mechanisms that contain the sample (fine mesh vs glass surface) might inherently be different physically and interact with the samples' properties. In field studies, the mesh screen samples are often associated with more dilution from ULW, as it samples a larger thickness than the glass plate. We do not expect this to have caused the differences in sampling efficiency during our experiment, as the ULW and SML should have been well-mixed. Finally, we did

not remove any data points, thereby assuming that the tank was well-mixed at all times, though Fig. A1 indicates otherwise. An extended QC would presumably change the results and likely decrease the variation, but potentially maintain a similar mean sampling efficiency. We conclude from this preliminary test, that our experimental design seems suited to repeat the sampling efficiency estimation for mesh screen sampling.

Figure A2 Sampling efficiency for DMS-d3 (left), isoprene-d5 (middle) and  $^{13}\text{CS}_2$  (right) in experiment A with the mesh screen in treatment fresh water (FW). Box width shows 25 to 75 percentile, centre line denotes the median, diamonds denote the mean, open circles denote outliers, and whiskers show minimum and maximum. The number of data points in experiment A is n = 13 (n = 10 for  $^{13}\text{CS}_2$ ).

885

# B Additional quality control using mass 91 applied in the context of Fig. Figure 3

Figure B1 Example from 8 August of linear fit between PA of DMS-d3 and mass 91 to identify poor-quality measurements. All samples are ULW samples. Linear fit excludes grey point.

Prior to assessing whether the tank was well-mixed (Sect. 3.2 Mixing in the tank), an additional step of quality control was performed on all ULW samples on each of the isotopes. *PA* of ULW samples that were far from the linear fit between *PA* of the respective isotope and mass 91 (e.g., grey point in Fig. B1) were flagged as poor-quality measurements and consequently removed from analysis. We assume that the poor-quality is caused by errors in measurement handling or short-term sensitivity drifts in the instrument. If the removed ULW sample was a reference sample, it was replaced with another ULW sample from the same sample set or, in the absence of more ULW samples, the sample set was discarded completely. SML samples formed a different cluster, due to the systematically lower PA compared to ULW (caused by decreased sampling efficiency) and, therefore, this step was not applied to SML samples. The sample indicated as being removed in Fig. B1 (mass spectrum ID "BULK\_13CM\_CC\_06") was the only ULW sample that was removed from the data set this way. It will be indicated with use\_for\_analysis\_flag=False in the published data set.

# C Effect of purging on surface activity measurements

Figure C1 Measured surface activity (SA) of purged and unpurged samples. ULW samples in treatment AS with surfactants (n = 30, green and black) are shown and a test sample set (n = 6, empty circles).

Due to the limited surface area of the tank used in experiment C ("AZ crate") two SML samples taken after each would not have been appropriate replicates of each other. Therefore, it was tested if the GC–MS sample could be reused for voltammetry analysis. The main differences for a reused GC–MS sample in voltammetry analysis are (1) longer standing time of sample before freezing and (2) being purged with helium for 10 min. These differences could lead to contamination due to the added handling. Figure C1 shows the data points used in the statistics of purged vs unpurged SA (Sect. 3.1.2 Surface activity and enrichment of surfactants). Only ULW samples were used, as we could safely assume that purging is the only source of error. Additionally, we took six subsamples (three for control, three were purged) from a beaker filled with ULW from the tank to specifically test the effect of purging. Surface activity is not affected by purging.

# D Results linear and multiple linear regressions in fresh water and artificial seawater from Sect. 3.4

Figure D1 Linear regressions of sampling efficiency for DMS-d3 (top), isoprene-d5 (middle) and  $^{13}$ CS<sub>2</sub> (bottom) against water temperature, salinity, spike volume per litre and number of dips in fresh water (FW) and artificial seawater (AS) treatment. Linear fit,  $R^2$  and number of samples (n) are given in top corner of each subplot.

Table D1 Multiple linear regression results for DMS-d3 in fresh water (FW) and artificial seawater (AS), using statsmodels smf.ols(...).fit(cov\_type='HC3')

| OI S | Regression   | Paculte |
|------|--------------|---------|
| ULS  | vent 622TOII | resutts |

|    | Dep. Variable:     | ratio_sml_over_ulw_d | msd3 | R-sq                                       | uared:     |        | 0.709   | )      |       |
|----|--------------------|----------------------|------|--------------------------------------------|------------|--------|---------|--------|-------|
|    | Model:             |                      | OLS  | Adj.                                       | R-squared: |        | 0.626   | 5      |       |
| 0  | Method:            | Least Squ            | ares | F-st                                       | atistic:   |        | 5.639   | )      |       |
|    | Date:              | Sat, 01 Nov          | 2025 | 5 Prob (F-statistic):<br>9 Log-Likelihood: |            |        | 0.00643 | 3      |       |
|    | Time:              | 15:3                 | 7:10 |                                            |            |        | 45.086  | 5      |       |
|    | No. Observations:  |                      | 19   | AIC:                                       |            |        | -80.17  | 1      |       |
|    | Df Residuals:      |                      | 14   | BIC:                                       |            |        | -75.45  | 5      |       |
| 5  | Df Model:          |                      | 4    |                                            |            |        |         |        |       |
|    | Covariance Type:   |                      | HC3  |                                            |            |        |         |        |       |
|    |                    |                      |      |                                            |            |        |         | [0.025 |       |
| 20 | Intercept          |                      | Θ.   | 1292                                       | 0.007      | 18.107 | 0.000   | 0.115  | 0.14  |
|    | temperature_water_ | _degc                | -0.  | 0223                                       | 0.011      | -2.021 | 0.043   | -0.044 | -0.00 |
|    | salinity_psu       |                      | 0.   | 0041                                       | 0.008      | 0.513  | 0.608   | -0.012 | 0.020 |
|    | cummulative_spike_ | _volume_mikrol_per_l | 0.   | 0342                                       | 0.012      | 2.769  | 0.006   | 0.010  | 0.05  |
| 25 | dips               |                      |      | 0117                                       |            | 1.350  |         | -0.005 | 0.02  |
| 23 | Omnibus:           | 0.354                |      | <br>in-Wat:                                |            |        | 1.951   |        |       |
|    | Prob(Omnibus):     | 0.838                | Jarq | ue-Ber                                     | a (JB):    |        | 0.074   |        |       |
|    | Skew:              | 0.145                | Prob | (JB):                                      |            |        | 0.964   |        |       |
|    | Kurtosis:          | 2.903                | Cond | . No.                                      |            |        | 2.58    |        |       |

## Notes:

[1] Standard Errors are heteroscedasticity robust (HC3)

Table D2 Multiple linear regression results for isoprene-d5 in fresh water (FW) and artificial seawater (AS), using statsmodels smf.ols(...).fit(cov\_type='HC3')

#### OLS Regression Results

|     | Dep. Variable: | ratio_sml_over_ulw_isoprened5 | R-squared:      | 0.691 |
|-----|----------------|-------------------------------|-----------------|-------|
| 940 | Model:         | 0LS                           | Adj. R-squared: | 0.602 |
|     | Method:        | Least Squares                 | F-statistic:    | 6.803 |

| Date:             | Sat, 01 Nov 2025 | <pre>Prob (F-statistic):</pre> | 0.00294 |
|-------------------|------------------|--------------------------------|---------|
| Time:             | 15:37:10         | Log-Likelihood:                | 43.711  |
| No. Observations: | 19               | AIC:                           | -77.42  |
| Df Residuals:     | 14               | BIC:                           | -72.70  |

Df Model: 4
Covariance Type: HC3

\_\_\_\_\_\_

|                                       | coet                                                                      | std err                                                                                                           | Z                                                                                                                                         | P> z                                                                                                                                                                 | [0.025                                                                                                                                                                                       | 0.975]                                                                                                                                                                                                                 |
|---------------------------------------|---------------------------------------------------------------------------|-------------------------------------------------------------------------------------------------------------------|-------------------------------------------------------------------------------------------------------------------------------------------|----------------------------------------------------------------------------------------------------------------------------------------------------------------------|----------------------------------------------------------------------------------------------------------------------------------------------------------------------------------------------|------------------------------------------------------------------------------------------------------------------------------------------------------------------------------------------------------------------------|
|                                       |                                                                           |                                                                                                                   |                                                                                                                                           |                                                                                                                                                                      |                                                                                                                                                                                              |                                                                                                                                                                                                                        |
| Intercept                             | 0.1267                                                                    | 0.008                                                                                                             | 16.425                                                                                                                                    | 0.000                                                                                                                                                                | 0.112                                                                                                                                                                                        | 0.142                                                                                                                                                                                                                  |
| temperature_water_degc                | -0.0169                                                                   | 0.012                                                                                                             | -1.455                                                                                                                                    | 0.146                                                                                                                                                                | -0.040                                                                                                                                                                                       | 0.006                                                                                                                                                                                                                  |
| salinity_psu                          | -0.0069                                                                   | 0.009                                                                                                             | -0.741                                                                                                                                    | 0.459                                                                                                                                                                | -0.025                                                                                                                                                                                       | 0.011                                                                                                                                                                                                                  |
| cummulative_spike_volume_mikrol_per_l | 0.0404                                                                    | 0.014                                                                                                             | 2.812                                                                                                                                     | 0.005                                                                                                                                                                | 0.012                                                                                                                                                                                        | 0.069                                                                                                                                                                                                                  |
| dips                                  | 0.0213                                                                    | 0.009                                                                                                             | 2.367                                                                                                                                     | 0.018                                                                                                                                                                | 0.004                                                                                                                                                                                        | 0.039                                                                                                                                                                                                                  |
|                                       | temperature_water_degc salinity_psu cummulative_spike_volume_mikrol_per_l | Intercept 0.1267 temperature_water_degc -0.0169 salinity_psu -0.0069 cummulative_spike_volume_mikrol_per_l 0.0404 | Intercept 0.1267 0.008 temperature_water_degc -0.0169 0.012 salinity_psu -0.0069 0.009 cummulative_spike_volume_mikrol_per_l 0.0404 0.014 | Intercept 0.1267 0.008 16.425 temperature_water_degc -0.0169 0.012 -1.455 salinity_psu -0.0069 0.009 -0.741 cummulative_spike_volume_mikrol_per_l 0.0404 0.014 2.812 | Intercept 0.1267 0.008 16.425 0.000 temperature_water_degc -0.0169 0.012 -1.455 0.146 salinity_psu -0.0069 0.009 -0.741 0.459 cummulative_spike_volume_mikrol_per_l 0.0404 0.014 2.812 0.005 | Intercept 0.1267 0.008 16.425 0.000 0.112 temperature_water_degc -0.0169 0.012 -1.455 0.146 -0.040 salinity_psu -0.0069 0.009 -0.741 0.459 -0.025 cummulative_spike_volume_mikrol_per_l 0.0404 0.014 2.812 0.005 0.012 |

\_\_\_\_\_\_

| Omnibus:       | 3.169  | Durbin-Watson:    | 1.648 |
|----------------|--------|-------------------|-------|
| Prob(Omnibus): | 0.205  | Jarque-Bera (JB): | 1.515 |
| Skew:          | -0.648 | Prob(JB):         | 0.469 |
| Kurtosis:      | 3.486  | Cond. No.         | 2.58  |

\_\_\_\_\_

Notes:

[1] Standard Errors are heteroscedasticity robust (HC3)

Table D3 Multiple linear regression results for  $^{13}CS_2$  in fresh water (FW) and artificial seawater (AS), using statsmodels smf.ols(...).fit(cov\_type='HC3')

#### OLS Regression Results

| 970 | Dep. Variable:    | ratio_sml_over_ulw_c13s2 | R-squared:       |        | 0.729   |                    |        |
|-----|-------------------|--------------------------|------------------|--------|---------|--------------------|--------|
|     | Model:            | OLS                      | Adj. R-squared:  |        | 0.652   |                    |        |
|     | Method:           | Least Squares            | F-statistic:     |        | 6.742   |                    |        |
|     | Date:             | Sat, 01 Nov 2025         | Prob (F-statist: | ic):   | 0.00305 |                    |        |
|     | Time:             | 15:37:10                 | Log-Likelihood:  |        | 45.959  |                    |        |
| 975 | No. Observations: | 19                       | AIC:             |        | -81.92  |                    |        |
|     | Df Residuals:     | 14                       | BIC:             |        | -77.20  |                    |        |
|     | Df Model:         | 4                        |                  |        |         |                    |        |
|     | Covariance Type:  | HC3                      |                  |        |         |                    |        |
| 980 |                   |                          | coef std err     | z      | P> z    | ========<br>[0.025 | 0.975] |
|     | Intercept         | 0.                       | 1202 0.007       | 17.774 | 0.000   | 0.107              | 0.133  |

|     | temperature_water_degc                  |        | -0.0219     | 0.010   | -2.104   | 0.035  | -0.042 | -0.001 |
|-----|-----------------------------------------|--------|-------------|---------|----------|--------|--------|--------|
|     | salinity_psu                            | 0.0053 | 0.007       | 0.758   | 0.448    | -0.008 | 0.019  |        |
| 985 | cummulative_spike_volume_mikrol_per_l   |        | 0.0353      | 0.012   | 2.932    | 0.003  | 0.012  | 0.059  |
|     | dips                                    |        | 0.0125      | 0.008   | 1.662    | 0.096  | -0.002 | 0.027  |
|     | ======================================= | ====== | :========   | ======= | ======== | ====   |        |        |
|     | Omnibus:                                | 0.054  | Durbin-Wats | on:     | ;        | 1.864  |        |        |
|     | Prob(Omnibus):                          | 0.973  | Jarque-Bera | (JB):   | (        | 0.147  |        |        |
| 990 | Skew:                                   | 0.094  | Prob(JB):   |         | 0.929    |        |        |        |
|     | Kurtosis:                               | 2.612  | Cond. No.   |         |          | 2.58   |        |        |
|     | ======================================= | ====== | =======     |         |          | ====   |        |        |

#### Notes:

[1] Standard Errors are heteroscedasticity robust (HC3)

## E Results linear and multiple linear regressions with artificial surfactants from Sect. 3.5

Figure E1 Linear regressions of sampling efficiency for DMS-d3 (top), isoprene-d5 (middle), and  $^{13}$ CS<sub>2</sub> (bottom) against water temperature, salinity, spike volume per litre and number of dips in fresh water (FW) and artificial seawater (AS) without and with surfactants treatment. Linear fit,  $R^2$  and number of samples (n) are given in top corner of each subplot. Linear regression of surface activity (SA) is shown in Fig. E2.

Figure E2 Linear regression of sampling efficiency for DMS-d3 (left), isoprene-d5 (middle), and  $^{13}$ CS<sub>2</sub> (right) against surface activity (SA) in SML in fresh water (FW) and artificial seawater (AS) without and with surfactants treatment. Linear fit,  $R^2$  and number of samples (n) are given in top corner of each subplot. Linear regressions of water temperature, salinity, spike volume per litre and number of dips are shown in Fig. E1.

Table E1 Multiple linear regression results for DMS-d3 in experiment C including treatments fresh water (FW), artificial seawater (AS) without and with surfactants, using statsmodels smf.ols(...).fit(cov\_type='HC3')

OLS Regression Results

| 1010 | ======================================= | ======================================= | ===== | =====    |             | ======= | ======= |         |          |
|------|-----------------------------------------|-----------------------------------------|-------|----------|-------------|---------|---------|---------|----------|
|      | Dep. Variable:                          | ratio_sml_over_ulw_o                    | dmsd3 | R-sq     | uared:      |         | 0.392   |         |          |
|      | Model:                                  |                                         | 0LS   | Adj.     | R-squared:  |         | 0.247   |         |          |
|      | Method:                                 | Least Sq                                | uares | F-st     | atistic:    |         | 2.006   |         |          |
|      | Date:                                   | Sat, 01 Nov                             | 2025  | Prob     | (F-statisti | c):     | 0.119   |         |          |
| 1015 | Time:                                   | 16:0                                    | 06:07 | Log-l    | Likelihood: |         | 62.118  |         |          |
|      | No. Observations:                       |                                         | 27    | AIC:     |             |         | -112.2  |         |          |
|      | Df Residuals:                           |                                         | 21    | BIC:     |             |         | -104.5  |         |          |
|      | Df Model:                               |                                         | 5     |          |             |         |         |         |          |
|      | Covariance Type:                        |                                         | HC3   |          |             |         |         |         |          |
| 1020 | ======================================= | ======================================= | ===== | =====    | =======     | ======= |         | ======= |          |
|      |                                         |                                         |       | coef     | std err     | _       |         | _       | 0.975]   |
|      | Intercept                               |                                         |       | <br>L393 | 0.006       | 23.395  | 0.000   | 0.128   | 0.151    |
|      | temperature_water_                      | degc                                    | -0.0  | 9324     | 0.016       | -1.995  | 0.046   | -0.064  | -0.001   |
| 1025 | salinity_psu                            |                                         | 0.0   | 9152     | 0.013       | 1.158   | 0.247   | -0.011  | 0.041    |
|      | cummulative_spike_                      | volume_mikrol_per_l                     | 0.0   | 9213     | 0.010       | 2.184   | 0.029   | 0.002   | 0.040    |
|      | dips                                    |                                         | -0.0  | 9016     | 0.007       | -0.242  | 0.809   | -0.015  | 0.011    |
|      |                                         |                                         |       |          |             |         |         |         |          |
|      | SA_mg_per_l_tx100_                      | eq_sml                                  | -0.0  | 9138     | 0.007       | -1.960  | 0.050   | -0.028  | -3.4e-06 |

Omnibus: 0.493 Durbin-Watson: 1.896 Prob(Omnibus): 0.781 Jarque-Bera (JB): 0.067 Skew: 0.115 Prob(JB): 0.967 Kurtosis: 3.081 Cond. No. 5.21

1035

Notes:

[1] Standard Errors are heteroscedasticity robust (HC3)

Table E2 Multiple linear regression results for isoprene-d5 in experiment C including treatments fresh water (FW) and artificial seawater (AS) without and with surfactants, using statsmodels smf.ols(...).fit(cov\_type='HC3')

#### OLS Regression Results

|      | ============      |                               |                     | ============ |
|------|-------------------|-------------------------------|---------------------|--------------|
|      | Dep. Variable:    | ratio_sml_over_ulw_isoprened5 | R-squared:          | 0.367        |
|      | Model:            | OLS                           | Adj. R-squared:     | 0.216        |
| 1045 | Method:           | Least Squares                 | F-statistic:        | 1.921        |
|      | Date:             | Sat, 01 Nov 2025              | Prob (F-statistic): | 0.133        |
|      | Time:             | 16:06:07                      | Log-Likelihood:     | 61.383       |
|      | No. Observations: | 27                            | AIC:                | -110.8       |
|      | Df Residuals:     | 21                            | BIC:                | -103.0       |
| 1050 | Df Model:         | 5                             |                     |              |
|      | Covariance Type:  | HC3                           |                     |              |

|      |                                       | coef    | std err | z      | P> z  | [0.025 | 0.975] |
|------|---------------------------------------|---------|---------|--------|-------|--------|--------|
| 1055 | Intercept                             | 0.1365  | 0.006   | 22.385 | 0.000 | 0.125  | 0.148  |
|      | temperature_water_degc                | -0.0309 | 0.016   | -1.993 | 0.046 | -0.061 | -0.001 |
|      | salinity_psu                          | 0.0169  | 0.013   | 1.265  | 0.206 | -0.009 | 0.043  |
|      | cummulative_spike_volume_mikrol_per_l | 0.0167  | 0.010   | 1.700  | 0.089 | -0.003 | 0.036  |
|      | dips                                  | 0.0016  | 0.007   | 0.231  | 0.817 | -0.012 | 0.015  |
| 1060 | SA_mg_per_l_tx100_eq_sml              | -0.0132 | 0.007   | -1.812 | 0.070 | -0.027 | 0.001  |

\_\_\_\_\_\_

0.000 Dumbin Waters

| ======================================= |        |                   | ========= |
|-----------------------------------------|--------|-------------------|-----------|
| Kurtosis:                               | 2.596  | Cond. No.         | 5.21      |
| Skew:                                   | -0.020 | Prob(JB):         | 0.911     |
| Prob(Omnibus):                          | 0.996  | Jarque-Bera (JB): | 0.186     |
| Umnibus:                                | 0.008  | Durbin-watson:    | 1.978     |

Notes:

[1] Standard Errors are heteroscedasticity robust (HC3)

1070

1065

 $Table~E3~Multiple~linear~regression~results~for~^{13}CS_2~in~experiment~C~including~treatments~fresh~water~(FW)~and~artificial~seawater~(AS)~without~and~with~surfactants,~using~statsmodels~smf.ols(...).fit(cov_type='HC3')$ 

| UI C | Regression | Docul | +- |
|------|------------|-------|----|

|      | ======================================= |                      | ===== | ======  |             |        | =======       |        |        |
|------|-----------------------------------------|----------------------|-------|---------|-------------|--------|---------------|--------|--------|
| 1075 | Dep. Variable:                          | ratio_sml_over_ulw_o | :13s2 | R−sqı   | uared:      |        | 0.377         |        |        |
|      | Model:                                  |                      | 0LS   | Adj.    | R-squared:  |        | 0.229         |        |        |
|      | Method:                                 | Least Squ            | ares  | F-sta   | atistic:    |        | 2.203         |        |        |
|      | Date:                                   | Sat, 01 Nov          | 2025  | Prob    | (F-statist  | ic):   | 0.0925        |        |        |
|      | Time:                                   | 16:6                 | 6:07  | Log-l   | _ikelihood: |        | 62.526        |        |        |
| 1080 | No. Observations:                       |                      | 27    | AIC:    |             |        | -113.1        |        |        |
|      | Df Residuals:                           |                      | 21    | BIC:    |             |        | -105.3        |        |        |
|      | Df Model:                               |                      | 5     |         |             |        |               |        |        |
|      | Covariance Type:                        |                      | HC3   |         |             |        |               |        |        |
| 1085 |                                         |                      |       | coef    | std err     | z      | P> z          | [0.025 | 0.975] |
|      | Intercept                               |                      |       | 1305    | 0.006       |        |               |        |        |
|      | temperature_water_                      | degc                 | -0.   | 0321    | 0.015       | -2.191 | 0.028         | -0.061 | -0.003 |
|      | salinity_psu                            |                      | Θ.    | 0212    | 0.012       | 1.836  | 0.066         | -0.001 | 0.044  |
| 1090 | cummulative_spike_                      | volume_mikrol_per_l  | Θ.    | 0159    | 0.010       | 1.534  | 0.125         | -0.004 | 0.036  |
|      | dips                                    |                      | -0.   | 0003    | 0.007       | -0.047 | 0.963         | -0.013 | 0.013  |
|      | <b>5</b> ·                              | eq_sml               |       |         |             | -2.045 |               | -0.027 | -0.001 |
|      | Omnibus:                                | <br>0.822            |       | in-Wats |             |        | ====<br>1.945 |        |        |
| 1095 | Prob(Omnibus):                          | 0.663                | Jarq  | ue-Bera | a (JB):     |        | 0.514         |        |        |
|      | Skew:                                   | 0.333                | Prob  | (JB):   |             |        | 0.774         |        |        |
|      | Kurtosis:                               | 2.882                |       | . No.   |             |        | 5.21          |        |        |

# 1100 Notes:

[1] Standard Errors are heteroscedasticity robust (HC3)

## F Calculations for diffusive boundary layer from Sect. 4.1.2

A simple box model was used to assess the effect of the diffusive boundary layer (DBL) on  $C_{SML}$ . A mass balance (Eq. (11) was set up for the complete tank with flux (Neumann boundary condition) only through the surface (i.e., no-flux Neumann boundary conditions on all other sides).

$$\frac{dn(t)}{dt} = -J_{surface}(t) \cdot A \tag{11}$$

where n is the moles as (absolute) PA (proportional to mol), t is time in s after mixing stopped, J is the flux in mol m<sup>-2</sup> s<sup>-1</sup> through the surface (defined as positive going inward), and A is the surface area of the tank in m<sup>2</sup>.

The flux through the surface is derived using Fick's first law of diffusion (Eq. (12)). This assumes that the water in the tank was quiescent,  $C_{air} \approx 0$  mol L<sup>-1</sup>, and the air is a perfect sink.

$$J_{surface}(t) = \frac{C_0}{\sqrt{\pi D t}} = \frac{n_0}{V_{tank}\sqrt{\pi D t}}$$
 (12)

where  $C_0 = n_0/V_{tank}$  is the initial concentration at t = 0 s in mol L<sup>-1</sup>, D is the diffusivity in m<sup>2</sup> s<sup>-1</sup>,  $n_0$  are the initial moles in mol,  $V_{tank} = AH$  is the volume of the water in the tank in m<sup>3</sup>, and H is the depth of the water in the tank in m.

The concentration in the ULW is used as initial condition, i.e.,  $C_0 = C_{ULW}$ .

Analytical solution of the ODE in Eq. (11) by integrating is given in Eq. (13).

$$n(t) = n_0 \exp\left(-\frac{2A}{V_{tank}} \sqrt{\frac{Dt}{\pi}}\right)$$
 (13)

where n(t) are the moles left in the tank at time t, and t is time in s after mixing stopped.

We calculate  $n_0$  from mean PA per experiment by multiplying with the respective volume of the tank. Water volume in experiment B was 34 cm × 78 cm × 26 cm, and 42 cm × 61 cm × 31 cm in experiment C. Diffusivity of DMS(-d3) ranged from 0.0000111–0.0000129 cm<sup>2</sup> s<sup>-1</sup> for the temperature range in this study. For the calculation,  $D = 1.29 \times 10^{-9}$  m<sup>2</sup> s<sup>-1</sup> (T = 20 °C,  $S_P = 0$ , calculated for DMS according to (Saltzman et al., 1993)). Due to missing diffusivity values for isoprene(-d5) and <sup>(13)</sup>CS<sub>2</sub>, these two are skipped in the calculation. As time we (conservatively) set t = 120 s, though the formation of a quasi-steady DBL only starts after the turbulent motion from the mixing has subsided, which is expected to take about 1 min. Finally, n(t = 120 s) yields that  $n(t)/n_0 = 99.9$  % of the moles are still left within the box (i.e., water volume) after 120 s.

To estimate the dilution from the DBL on the glass plate, we partition this n(t = 120 s) between ULW and DBL. Under the assumption that within this short time  $C_{ULW}$  did not change, i.e.,  $C_{ULW} = C_0$ , we calculate how many moles of n(t) are present in the DBL with Eq. (14).

$$n_{DBL} = n(t) - C_0 V_{ULW} \tag{14}$$

where  $n_{DBL}$  are the moles in the DBL,  $V_{ULW} = A (H - h_{DBL})$  is the volume of the water below the DBL in the tank in m<sup>3</sup>, and  $h_{DBL}$  is the thickness of the DBL in m, approximated with Eq. (15).

$$h_{DBL} = a\sqrt{\pi Dt} \tag{15}$$

where  $h_{DBL}$  is the thickness of the DBL in m, a = 3 is a common coefficient that defines  $h_{DBL}$  in relation to ~99 % of  $C_{ULW}$ . 1130 Finally, the concentration in the DBL in relation to the concentration in the ULW, i.e. the concentration sampled by the glass plate is calculated by Eq. (16).

$$\frac{C_{DBL}}{C_{ULW}} = \frac{n_{DBL}}{V_{DBL}C_{ULW}} \tag{16}$$

where  $C_{DBL}$  is the concentration in the DBL,  $C_{ULW}$  is the concentration in the ULW in mol L<sup>-1</sup>, and  $V_{DBL} = A h_{DBL}$  is the volume of the water in the DBL in m<sup>3</sup>, which equates to 0.79 for both experiments.