# Peer review of "Glass plate sampling efficiency for trace gases in the sea surface microlayer"

_EGUsphere, 2025_

## Referee Comment (RC2)

This manuscript addresses an important topic, trace gas sampling in the Surface Microlayer (SML), through a systematic laboratory approach with the aim to characterize sampling losses when the SML is sampled using a glass plate. The manuscript is generally well written and the authors have well-constructed their argumentation and reasoning. However, the manuscript also demonstrates that the mechanisms regulating air-sea exchange are very complex and not fully understood yet. Therefore, several limitations prevent a wide application of the evidenced sampling efficiencies.

In particular the influence of surfactants is complex and the choice of one surfactant is limiting the general representativeness of the sampling efficiencies, particularly in cases where surfactants are present (as is generally the case in natural environments).

There seems to be a known and recognised bias in the experimental design where the oversaturation of the trace gas in the aqueous phase and zero-air above, will introduce a very strong gradient and may not uphold the well-mixed assumption. While this is discussed in the manuscript, this should be mentioned earlier on, e.g. in the experimental set-up, to better frame the limitations of the sampling efficiencies that are found.

While the manuscript clearly motivates the use of the peak area and not the concentrations of the gases, it would be good to indicate at least the LoD for each gas and the system's sensitivity for each. This will help interpret the magnitude differences for PAs and their ratios discussed and give a better estimation of the environmental relevance of the method. Indeed, without these information, the differences in PA between trace gases explained as different amounts of trace gas in the sample, could also be due to different trapping and ionisation efficiencies between these very different gasses (as for example shown also in Wang et al., 2023, DOI: 10.1016/j.marchem.2023.104206 ) Therefore, the information on how sensitive the purge and trap GC-MS system is, seems necessary.

Although the influence of the surfactant on the sampling efficiency seem not very high, it remains a relatively badly understood system. The use of the surface activity might be a convenient choice, but the SA might not be the best indicator for the exchange of gases, as it will also highly depend on the nature of the surfactant, which this method does not qualify. Different (mixes) of surfactants will behave differently towards different gas species, depending on their physicochemical nature. As is mentioned in the conclusion, the influence of the surfactants needs to be further studied, but not only with regard to their concentration (which was attempted here) but also their nature (ionic, nonionic). It would be nice to add this nuance in the text, or in the conclusion.

Another suggestion for an additional nuance in the conclusion, would be to consider a non-linear approach to this dataset (or an extended) as well seen the multiple issues with linear assumptions that are not validated and the poor modelling with surfactants.

Further minor comments:

L126-127: The text states that 'Two different tanks were used… ', however table 1 lists 3 different tanks. The incubation bath should also briefly be described in the Experimental setup (2.1)

L306: It is mentioned that on 3 occasions the EF was below 1; it is not clear to me if these values were included in table 2? Do the authors have an idea why these samples are different (First of a set? Temperature?...)

L524: '…dilution by the DBL reduced the PA sampled by the glass plate to about 0.79 of the concentration in the ULW…' : Please clarify what are the units of this 0.79 as I first assumed this was percentage or a ratio but now think it might be absolute reduction in a.u. of the peak area?

Tables and Figures:

Figure 2: difference between different sample types is difficult to see due to small symbols

Table 6: what is the meaning of the asterisks (*) in the table?

For all figure references: in the text is mentioned 'Fig. Figure x' ; this should normally just read 'Figure x'.

Language and editorial :

L77: it's not clear to me what is meant by undersize here. Could you perhaps find a synonym?

L107 : … experiment A *is* described in Appendix A…  please correct

L306: … was enriched on average EF from 3.7… this sentence does not seem grammatically correct. Please correct.

L625: …. expectations and findings by : this sentence seems not grammatically correct , please correct.

---

## Author Comment (AC2)

**Lange et al.—Reply to RC2**

RC2: 'Comment on egusphere-2025-5361', Anonymous Referee #2, 07 Jan 2026

Review EGUsphere Lange et al.

This manuscript addresses an important topic, trace gas sampling in the Surface Microlayer (SML), through a systematic laboratory approach with the aim to characterize sampling losses when the SML is sampled using a glass plate. The manuscript is generally well written and the authors have well-constructed their argumentation and reasoning. However, the manuscript also demonstrates that the mechanisms regulating air-sea exchange are very complex and not fully understood yet. Therefore, several limitations prevent a wide application of the evidenced sampling efficiencies.

In particular the influence of surfactants is complex and the choice of one surfactant is limiting the general representativeness of the sampling efficiencies, particularly in cases where surfactants are present (as is generally the case in natural environments).

Thanks for the thorough comment! We observe much overlap between RC1 and RC2 with respect to the points addressed and also the indications on how to resolve them, which strengthens the arguments given in both reviews. We will indicate which of our replies were similar in RC1. We will especially rework the *Abstract* and *Section 5 Conclusion and outlook* in view of the comments made in this review. We carefully revised our manuscript with the suggestions given and will change it as follows.

There seems to be a known and recognised bias in the experimental design where the oversaturation of the trace gas in the aqueous phase and zero-air above, will introduce a very strong gradient and may not uphold the well-mixed assumption. While this is discussed in the manuscript, this should be mentioned earlier on, e.g. in the experimental set-up, to better frame the limitations of the sampling efficiencies that are found.

We will add the information about oversaturation in line 123ff (*Section 2.1 Experimental setup*), e.g. ", but in all cases the trace gases in the water were oversaturated, causing a steep gradient towards the near-zero atmosphere.". We will add "to homogenize any gradients that had formed due to the oversaturation in the water." behind "we mixed the water body carefully for one min" in line 133 (*Section 2.2 Sampling*). Additionally, we will revise *Section 2 Methods* to include the information.

While the manuscript clearly motivates the use of the peak area and not the concentrations of the gases, it would be good to indicate at least the LoD for each gas and the system's sensitivity for each. This will help interpret the magnitude differences for PAs and their ratios discussed and give a better estimation of the environmental relevance of the method. Indeed, without these information, the differences in PA between trace gases explained as different amounts of trace gas in the sample, could also be due to different trapping and ionisation efficiencies

between these very different gasses (as for example shown also in Wang et al., 2023, DOI: 10.1016/j.marchem.2023.104206 )Therefore, the information on how sensitive the purge and trap GC-MS system is, seems necessary.

We will add (exemplary) LOD and sensitivity in mol $L^{-1}$ and slope (PA per mol $L^{-1}$) in *Supplement S1* for the presented calibration curves, where also the linearity is presented (as requested in RC1). We will discuss that the differences in PA between the trace gases cannot be compared. We will add that the standard was prepared in different concentration ranges for each of the gases in line 123.

Although the influence of the surfactant on the sampling efficiency seem not very high, it remains a relatively badly understood system. The use of the surface activity might be a convenient choice, but the SA might not be the best indicator for the exchange of gases, as it will also highly depend on the nature of the surfactant, which this method does not qualify. Different (mixes) of surfactants will behave differently towards different gas species, depending on their physicochemical nature. As is mentioned in the conclusion, the influence of the surfactants needs to be further studied, but not only with regard to their concentration (which was attempted here) but also their nature (ionic, nonionic). It would be nice to add this nuance in the text, or in the conclusion.

We agree now, with hindsight, that the surface activity is insufficient when addressing the influence of surfactants on trace gas properties in the medium. Unfortunately, we cannot fix this problem for this study, but we will address this as relevant for future studies. We will add the nature of surfactants (soluble/insoluble, ionic/non-ionic, groups like lipids/amino acids/TEP) as potential drivers to nuance better our conclusion in *Section 4.1.3 Influence of surfactants addition* (*Discussion*). We will also address in the *Discussion* that the added surfactant does not represent the complex mixture of natural surfactants. The latter was also requested in RC1. Comprehensive definition of a model surfactant mix are lacking, but a potential second phase of the BASS project will study this particular question.

Another suggestion for an additional nuance in the conclusion, would be to consider a non-linear approach to this dataset (or an extended) as well seen by? the multiple issues with linear assumptions that are not validated and the poor modelling with surfactants.

We will add in *Section 5 Conclusion and outlook* that it was not possible to perform a non-linear assessment of the results, as the experimental setting was too constrained. We will also indicate in line 686ff that the extension to a broader range of environmental factors should be accompanied by non-linear and more complex analysis.

Further minor comments:

L126-127: The text states that 'Two different tanks were used… ', however table 1 lists 3 different tanks. The incubation bath should also briefly be described in the Experimental setup (2.1)

We agree that this is inconsistent usage. We will add a footnote in Table 1 that experiment A is further described in *Appendix A*.

L306: It is mentioned that on 3 occasions the EF was below 1; it is not clear to me if these values were included in table 2? Do the authors have an idea why these samples are different (First of a set? Temperature?...)

We will rephrase line 306ff "(with three exceptions where EF < 1.0, once for each level of TX-100 added)" to clarify that these three exceptions are not average values and that they are part of the averages presented in Table 2.

In the analysis, we checked whether the EF<1 was related to sample sequence (i.e., first sample of day or after a long break), how often the tank had been mixed before already, (absolute) surface activity, amount of Triton X-100 added, age of the water (i.e., time since filling), contamination from sampling equipment, temperature, salinity, and reported deviations from standard procedure in sampling protocols (e.g., person sampling, mixing times, unexpected events), GC-MS measurement protocols (because each glass plate sample was first measured for trace gas content) and voltammetry measurement protocols (e.g., handling errors, failures). There was no indication that any of those were causing the EF<1. Therefore, we did not exclude them from our analysis. For ULW samples, we took triplicates and therefore had a means to identify bad replicates, while for the glass plate samples they were all singlets, though each sample was measured several times with voltammetry. Nonetheless, we cannot quantify whether the EF<1 sample pairs are accurate or if something was off with the glass plate sample.

L524: '…dilution by the DBL reduced the PA sampled by the glass plate to about 0.79 of the concentration in the ULW…' : Please clarify what are the units of this 0.79 as I first assumed this was percentage or a ratio but now think it might be absolute reduction in a.u. of the peak area?

It is a ratio. We will rephrase line 532ff to remove the confusion.

Tables and Figures:

Figure 2: difference between different sample types is difficult to see due to small symbols
We will increase symbol size.

Table 6: what is the meaning of the asterisks (*) in the table?
The info is missing (significance level). We will add the explanation of the asterisk in the caption.

For all figure references: in the text is mentioned 'Fig. Figure x' ; this should normally just read 'Figure x'.
Thank you for pointing out. We will correct this issue in the revised manuscript. Same suggestion made in RC1.

Language and editorial:

L77: it's not clear to me what is meant by undersize here. Could you perhaps find a synonym?
It refers to the effect that particles can be too small to be captured by a technique. This is more relevant for organic material than trace gases actually. Therefore, in this context we will remove the word from the list to avoid confusion.

L107 : … experiment A is described in Appendix A…  please correct
Done.

L306: … was enriched on average EF from 3.7… this sentence does not seem grammatically correct. Please correct.
Done.

L625: …. expectations and findings by: this sentence seems not grammatically correct, please correct.
Done.